# Targeting chondrocytes for arresting bony fusion in ankylosing spondylitis

Fenli Shao[1,9], Qianqian Liu[1,9], Yuyu Zhu [1,2,9], Zhidan Fan[3,9], Wenjun Chen[4], Shijia Liu[4], Xiaohui Li[5], Wenjie Guo[1], Gen-Sheng Feng[6], Haiguo Yu[3✉], Qiang Xu[1,7✉] & Yang Sun [1,7,8✉]

Bony fusion caused by pathological new bone formation manifests the clinical feature of ankylosing spondylitis (AS). However, the underlying mechanism remains elusive. Here we discovered spontaneous kyphosis, arthritis and bony fusion in mature CD4-Cre;Ptpn11[f/f] mice, which present the pathophysiological features of AS. A population of CD4-Cre-expressing proliferating chondrocytes was SHP2 deficient, which could differentiate into pre-hypertrophic and hypertrophic chondrocytes. Functionally, SHP2 deficiency in chondrocytes impeded the fusion of epiphyseal plate and promoted chondrogenesis in joint cavity and enthesis. Mechanistically, aberrant chondrocytes promoted ectopic new bone formation through BMP6/pSmad1/5 signaling. It is worth emphasizing that such pathological thickness of growth plates was evident in adolescent humans with enthesitis-related arthritis, which could progress to AS in adulthood. Targeting dysfunctional chondrogenesis with Smo inhibitor sonidegib significantly alleviated the AS-like bone disease in mice. These findings suggest that blockade of chondrogenesis by sonidegib would be a drug repurposing strategy for AS treatment.

[1] State Key Laboratory of Pharmaceutical Biotechnology, Department of Biotechnology and Pharmaceutical Sciences, School of Life Sciences, Nanjing University, 163 Xianlin Avenue, Nanjing 210023, China. [2] College of Pharmacy, Nanjing University of Chinese Medicine, 138 Xianlin Avenue, Nanjing 210023, China. [3] Department of Rheumatology and Immunology, Children's Hospital of Nanjing Medical University, 72 Guangzhou Road, Nanjing 210008, China. [4] Affiliated Hospital of Nanjing University of Chinese Medicine, 155 Hanzhong Road, Nanjing 210029, China. [5] Department of Radiology, Children's Hospital of Nanjing Medical University, 72 Guangzhou Road, Nanjing 210008, China. [6] Department of Pathology, and Division of Biological Sciences, University of California San Diego, La Jolla, CA 92093, USA. [7] Jiangsu Key Laboratory of New Drug Research and Clinical Pharmacy, Xuzhou Medical University, 209 Tongshan Road, Xuzhou 221004, China. [8] Chemistry and Biomedicine Innovation Center (ChemBIC), Nanjing University, 163 Xianlin Avenue, Nanjing 210023, China. [9]These authors contributed equally: Fenli Shao, Qianqian Liu, Yuyu Zhu, Zhidan Fan. ✉email: yangsun@nju.edu.cn; qiangxu@nju.edu.cn; haiguoyu2000@163.com

Ankylosing spondylitis (AS) is a chronic seronegative arthritis that primarily affects the axial skeleton[1]. It is highly heritable (heritability > 90%) and has a strong association with human leukocyte antigen-B27 gene[2,3]. Besides inflammatory back pain and arthritic destruction, pathological new bone formation is a predominant feature of AS, which is very different from clinical features of rheumatoid arthritis[4]. When the osteophytes bridge the entire joint cavity, the affected joint will be immobilized, resulting in axial joint stiffness, spinal ankyloses, and even permanent disability. Although existing treatments reduce inflammation effectively in active AS, including non-steroidal anti-inflammatory drugs (NSAIDs), TNF-α antibody and interleukin-17 inhibitors, these treatments cannot significantly alleviate the radiographic progression in AS[5–7].

The mechanism of pathological new bone formation remains elusive at present. MRI results showed that new bone formation in vertebral syndesmophytes is more likely to occur in the sites with previous inflammation than in the area with no inflammation[8]. In addition, inflammation promoted expression of osteoinductive proteins and facilitated pathological new bone formation at the enthesis indirectly[9,10]. Depending on the hypothesis that inflammation activates the repair process followed by new bone formation, the disability of anti-inflammation treatments would be attributable to that it is too late to prevent the formation of ectopic new bone in patients with AS, who conforms to the diagnostic criteria of enthesitis and definite radiographic sacroiliitis. Moreover, the majority of new vertebral syndesmophytes develop at sites with no evidence of previous inflammation[11,12], suggesting new bone formation in AS is not merely induced by inflammation.

The osteophytes are formed by endochondral ossification, which is accomplished by the differentiation of periosteal cells to chondroblasts and osteoblasts[13]. Recently, chondrocyte differentiation was observed in the ligaments of patients with early-stage AS, and cartilage formation was followed by calcification[14]. Along with bony fusion, the prevalence rate of osteoporosis is 9.5–40% in AS[15,16]. Increased osteoclasts were found on the bony surfaces of calcified cartilage and subchondral bone marrow in AS[14,17]. It suggests chondrogenesis and osteoclastogenesis participate in the progression of AS.

SHP2 (encoded by Ptpn11) is a ubiquitously expressed protein that regulates cell survival, proliferation, and differentiation[18]. In our previous work, we demonstrated that SHP2 was involved in the pathogenesis of a variety of inflammation-related diseases[19–21]. Conditional deletion of SHP2 in different subsets of chondrocytes results in different phenotypes, including exostoses, dwarfism, and decreased bone mineralization[22–25]. In chondroid progenitors, SHP2 deficiency increased the expression of Indian hedgehog (Ihh) and parathyroid hormone-related protein (Pthrp) and caused metachondromatosis, which could be ameliorated by the treatment of smoothened (Smo) inhibitor[22,26]. Sonidegib is an oral smoothened antagonist that blocks hedgehog signaling and is used in the therapy of basal cell carcinoma[27]. It is acknowledged that BMP proteins, Wnt signaling pathway, and hedgehog proteins direct the process of osteogenesis in AS[1,28], but whether blocking these pathways could retard the bony fusion in AS is unknown.

In the present study, we serendipitously discovered a spontaneous AS-like bone disease in mature CD4-Cre;Ptpn11f/f mice, which show symptoms of kyphosis, scoliosis, arthritis, and bony fusion of axial joints. Lineage tracing and functional analysis revealed that CD4-Cre mediated SHP2 deficiency in pre-hypertrophic and hypertrophic chondrocytes, which disturbed the fusion of epiphyses (cartilaginous endplate and growth plate). Aberrant chondrocytes induced structure damage and ectopic new bone formation through endochondral ossification in vivo and in vitro, mediated at least in part via BMP6/pSmad1/5 signaling. Smoothened (Smo) inhibitor sonidegib abolished

chondrogenesis and significantly ameliorated the AS-like bone disease in CD4-CKO mice. These findings indicate that targeting chondrocytes is a promising strategy for arresting the radiological progression in AS.

## Results

**Ptpn11 deletion in CD4-expressing cells causes age-related AS-like bone disease.** CD4-Cre;Ptpn11f/f (CD4-CKO) mice were generated by crossing Ptpn11f/f (CD4-Ctrl) mice with mice expressing Cre driven by endogenous CD4 promoter. The CD4-CKO mice developed bone disease spontaneously, which started with enlarged pelvic incidence angle and knee stiffness at 7 months, followed by kyphosis and ankylosis of hip and knee joint (Fig. 1a, b, Supplementary Fig. 1a, and Supplementary Movie 1). In aged CD4-CKO mice, structure damage and bony fusion led to permanent disability and increased the mortality rate of mice (Supplementary Fig. 1b). Interestingly, neither CD4-Ctrl nor CD4-cre;Ptpn11f/+ mice displayed a discernible phenotype. Radiography and skeletons images showed bony fusion and osteophyte focused on the spine, wrist, and sacroiliac, hip and knee joints rather than ankles and toes (Fig. 1c, d, Supplementary Fig. 1c, d, Supplementary Fig. 2, and Supplementary Fig. 3). Micro-CT imaging revealed slight bone deformation and osteophytes in spine and knee joint of 7-month-old mature CD4-CKO mice, followed by massive osteophytes, bony fusion, and anky-losis in hip joint, knee joint, sacroiliac joint, and spine of 12-month-old aged CD4-CKO mice (Fig. 1e–h).

Osteoporosis (or low bone mineral density) is a frequent complication of AS, which increases fracture risk[15]. The femoral bone mineral density (BMD) was significantly lower in CD4-CKO mice than CD4-Ctrl littermates, although trabecular bone mass did not decrease (Fig. 1i, Supplementary Fig. 4a, b). This decreased BMD is likely attributed to the severe loss of compact bone and subchondral bone (red arrow), which led to increased bone fracture risk (Fig. 1j, Supplementary Fig. 4c). The bone disease in CD4-CKO mice was analogous to the radiological disorders in patients with AS, which is characterized by structure damage and bony fusion in axial joints, including spine, sacroiliac, and knee joints (Fig. 1k).

**Ectopic new bone formation is accompanied by aberrant chondrocytes and inflammation in CD4-CKO mice.** Bone remodeling is a balance between bone formation and bone resorption regulated by osteoblasts and osteoclasts, respectively[29]. Tartrate-resistant acid phosphatase (TRAP) staining for osteo-clasts revealed more osteoclasts in mature and aged CD4-CKO mice than CD4-Ctrl littermates (Fig. 2a). However, the osteoblast numbers were comparable between CD4-CKO mice and CD4-Ctrl littermates (Fig. 2b). Therefore, increased bone resorption led to osteoporosis and increased risk of bone fracture in CD4-CKO mice.

H&E and SOFG staining revealed degeneration of articular cartilage (black arrows) and ectopic new bone formation (blue arrows) in mature and aged CD4-CKO mice (Fig. 2c, d). Cartilage abnormality and bone deformation were also detected in the spine, hip joints, and wrist in aged CD4-CKO mice (Fig. 2e–g, Supplementary Fig. 5a). In epiphyseal plate, chondrocytes assume proliferating, pre-hypertrophic and hypertrophic phenotypes, and mediated endochondral ossification[30]. The epiphyseal plate of mature CD4-CKO mice was apparently thicker than that of CD4-Ctrl littermates (Fig. 2h, i), suggesting abnormal endochondral ossification in mature CD4-CKO mice. Moreover, CD4-CKO mice showed enthesitis and synovial pannus in bone lesions (Fig. 2j, Supplementary Fig. 5b, c). ELISA analysis revealed the levels of TNF-α, IL-6, IL-17A, and CXCL10 were increased in serum of CD4-CKO mice (Supplementary Fig. 6). These results

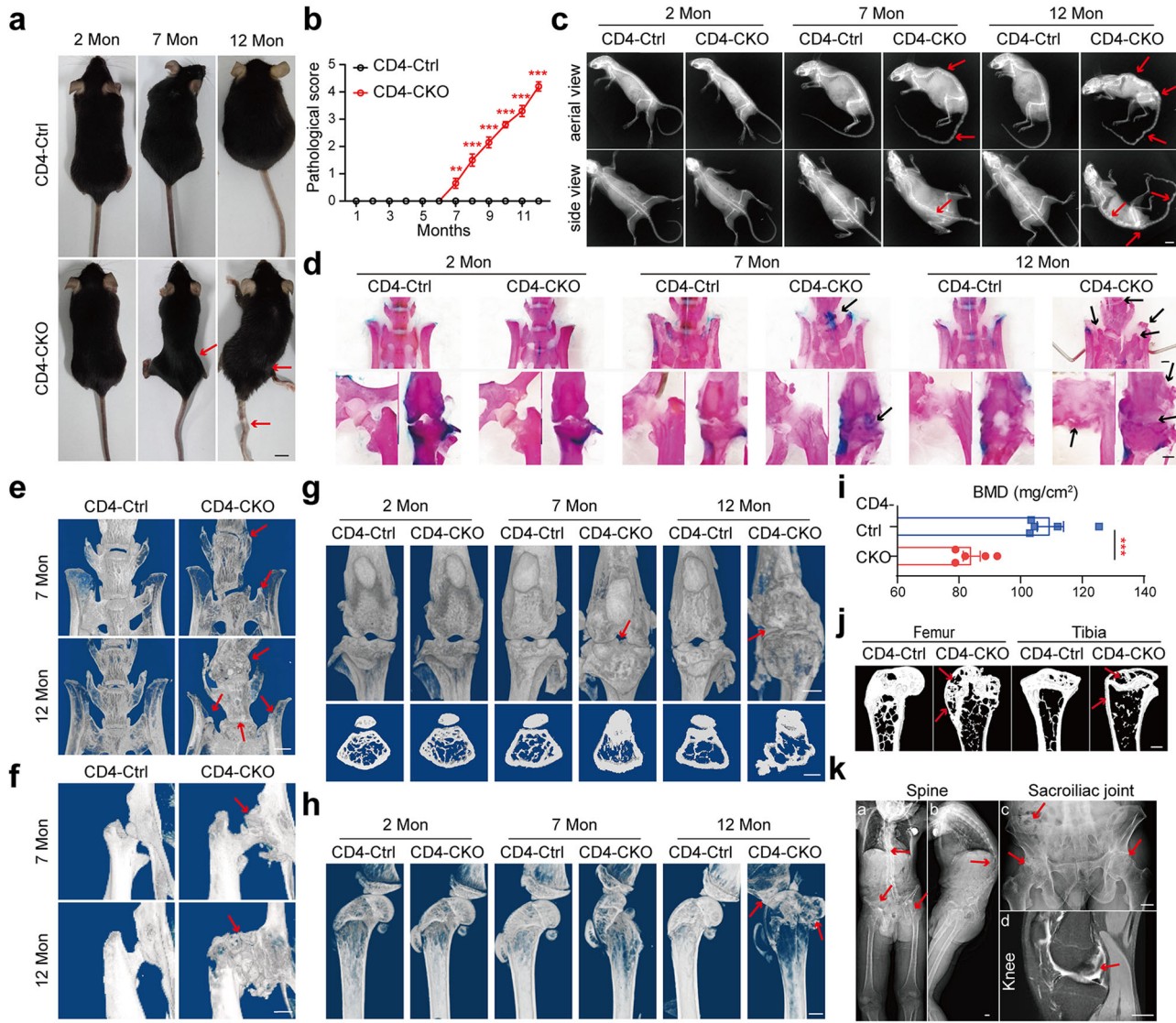

**Fig. 1 Ptpn11 deletion in CD4-expressing cells causes age-related AS-like bone disease. a** Gross images of female mice. **b** Pathological scores ($n = 10$). **c** X-ray images of female mice. Red arrows show kyphosis and joint ankylosis in mature and aged CD4-CKO mice. **a, c** Scale bars: 1 cm. **d** Skeleton staining results indicate increased Alcian blue staining (cartilage) in mature CD4-CKO mice and bone fusion in aged CD4-CKO mice. Black arrows show bone deformation. **e–h** μ-CT radiographs show bone structure of sacroiliac joints, spines (**e**), hip joints, (**f**) and knee (**g, h**). Arrows show osteophytes and bony fusion in spine, hip joints and knee joints. **i** Femoral bone mineral density (BMD) ($n = 5$) and **j** Sagittal sections of femur and tibia show decreased BMD and loss of cortical bone and subchondral bone in aged female CD4-CKO mice. Arrows show bone loss. **d–h, j** Scale bars: 1 mm. **k** Radiological images of patients with AS. Arrows indicate outward acetabular labrum, ankylosis of hip and sacroiliac joints and abnormity of epiphysis respectively. Scale bars: 2.5 cm. **b, i** Data are presented as mean ± SEM. **b** $**p < 0.01$, $***p < 0.001$, **i** $p$-value $= 0.0010$, determined by two-tailed Student's $t$-test. **a, c–h, j, k** Data are representative of three independent biological replicates.

reveal the ectopic new bone formation is coincident with enthesitis and aberrant chondrocytes in CD4-CKO mice.

**Immune system abnormality is not the cause of AS-like bone deformation.** CD4 is a membrane glycoprotein expressed on helper T cells. Thus, we hypothesized that SHP2-deficient T cells disturb skeletal balance in CD4-CKO mice. Although the proportions of Th1, Th17, Treg and CD8[+]IFN-γ[+] cells were significantly higher in aged CD4-CKO mice, the proportions of T-cell subsets in inguinal lymph nodes of young and mature CD4-CKO mice were comparable to those of the CD4-Ctrl littermates (Fig. 3a, Supplementary Fig. 7). Furthermore, the anti-rheumatic drug Iguratimod, which has potent anti-inflammatory activity[31], offered no protection against bone deformation in

CD4-CKO mice (Supplementary Fig. 8). To explore whether SHP2-deficient T cell is critical for AS-like bone disease, CD3[+] T cells were isolated from the spleens of CD4-Ctrl mice (WT T cells) and CD4-CKO mice (KO T cells) and transferred into immunodeficient NCG mice, respectively[32]. Flow cytometry analysis showed CD3[+] T cells were detected in the recipient's spleens after transfer for 2 months (Fig. 3b). However, the recipient mice in neither group showed a bone abnormality, despite the obvious inflammation in the skin of 12-month-old recipients with KO T cells (Fig. 3c, Supplementary Fig. 9a, b). Furthermore, we created Lck-Cre;Ptpn11[f/f] (Lck-CKO) mice, in which SHP2 deletion was driven by the expression of the lymphocyte protein tyrosine kinase (Lck) gene in T cells. Gross image and X-ray analysis showed identical skeleton and femoral BMD in Lck-CKO mice and their WT littermates (Fig. 3d–f). These

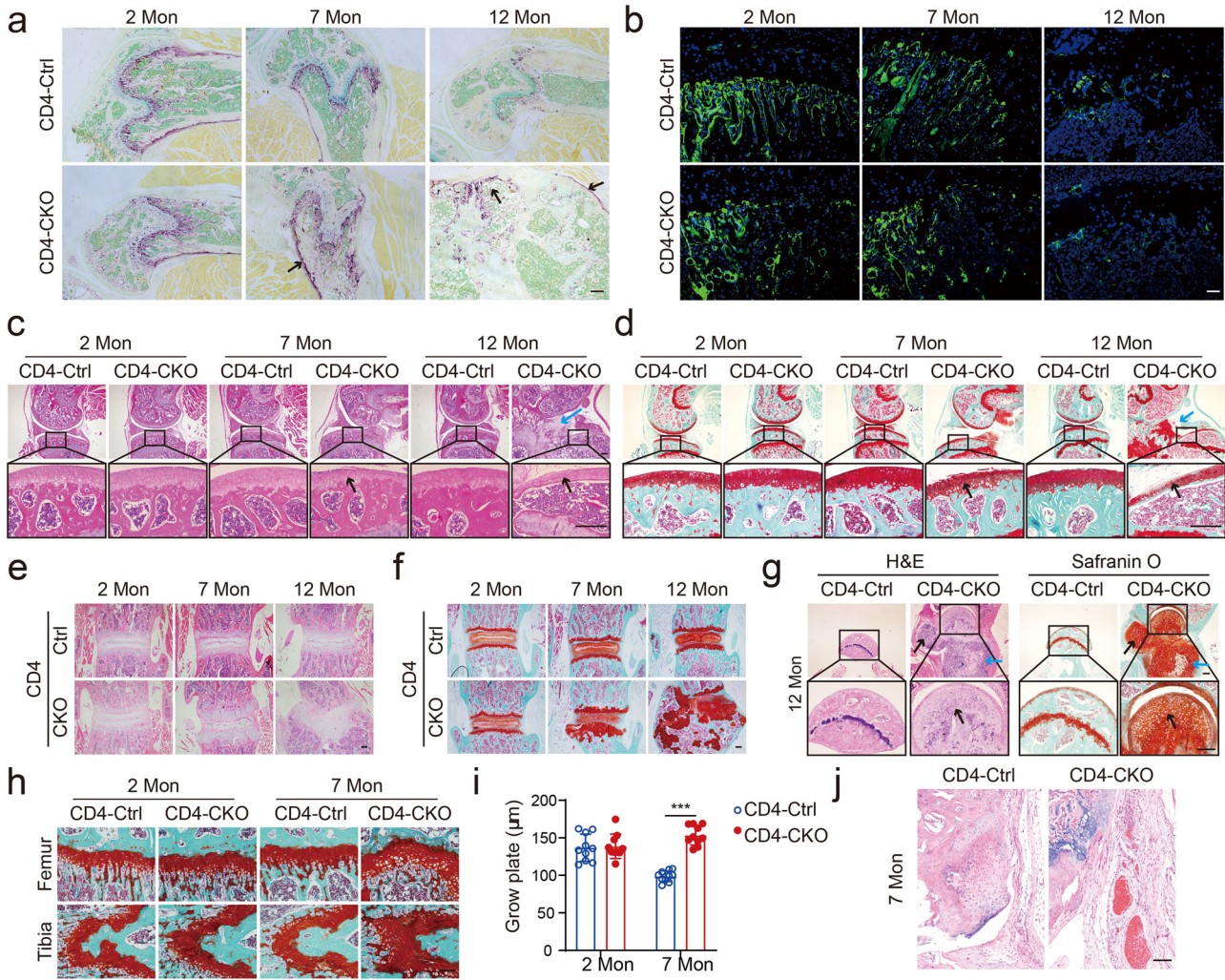

**Fig. 2 Ectopic new bone formation is accompanied by aberrant chondrocytes and inflammation in CD4-CKO mice. a** TRAP staining of distal femur stained. **b** Osteocalcin immunostaining images of tibia sections. Arrows indicate osteoclasts. **c**, **d** H&E and Safranin O-Fast Green (SOFG) analysis of knee joints. Black arrows show degenerated articular cartilage, and blue arrows show ectopic new bone formation in articular cavity and enthesis. **e**, **f** H&E and SOFG staining images of spine section. **g** H&E and SOFG staining images of hip joint section. **a–g** Scale bars: 200 μm. **c**, **d**, **g** Black arrows show degenerated articular cartilage, and blue arrows show ectopic new bone formation in articular cavity and enthesis. Data are representative of three independent biological replicates. **h**, **i** SOFG staining images and measure of the epiphyseal plate. Scale bars: 50 μm. **i** Data are presented as mean ± SEM ($n = 10$). ***$p < 0.001$, determined by two-tailed Student's $t$-test. **j** H&E staining images show inflammation, pannus formation, and proliferating synoviocytes in knee joints. Scale bars: 100 μm. Data are representative of five independent biological replicates.

data indicate SHP2-deficient T cells cannot induce AS-like bone lesions independently.

To explore the cause of bone lesions, we created CD4-CKO;Rosa26-mTmG mice by crossing CD4-CKO mice with double-fluorescent Cre reporter Rosa26-mTmG mice[33]. In CD4-CKO;Rosa26-mTmG mice, flow cytometry and immunostaining results showed that the T cells, as well as CD45+CD11b+Ly6G+ neutrophils, were GFP+ cells, indicating these immune cells were SHP2 deficiency in CD4-CKO mice (Supplementary Fig. 9c–e). Then, we performed a bone marrow transplant, in which the CD4-Ctrl and CD4-CKO mice as the donors and recipients respectively. As shown in Fig. 3g, h, bone disease actually developed in irradiated CD4-CKO mice that received CD4-Ctrl bone marrow cells (WT-KO) rather than irradiated CD4-Ctrl mice that received CD4-CKO bone marrow cells (KO-WT) (Fig. 3g, h, Supplementary Movie 2). To our interest, the bone deformation in WT-KO mice was almost equal to the bone disease in irradiated CD4-CKO mice that received CD4-CKO bone marrow cells (KO-KO). Radiographic and histological

images showed the structure damage and ectopic new bone formation in CD4-CKO chimeras (Fig. 3i–k). These data suggest SHP2 deficiency in the immune system is dispensable for AS-like bone disease in CD4-CKO mice.

**CD4-Cre mediates conditional deletion of SHP2 in chondrocytes.** CD4-Cre-mediated conditional *Sos1/2* or *Erk1/2* deletion in chondrocytes resulted in cartilage tumors[34–36]. In addition, it has been reported that CD4-Cre mediated *Ptpn11* deletion in some chondrocytes and led to cartilage tumors in wrist bones[37]. Then, we performed the immunohistologic analysis in the knee joint of CD4-CKO mice. The immunostaining images showed Cre expressed in proliferating chondrocytes in inner layer of articular cartilage and proliferating zone of the growth plate (Supplementary Fig. 10). Immunofluorescence analysis showed GFP+ chondrocytes were concentrated at the center of the cartilage mold (blue dotted box) and pre-hypertrophic and hypertrophic zone of the growth plate (green dotted box), which were

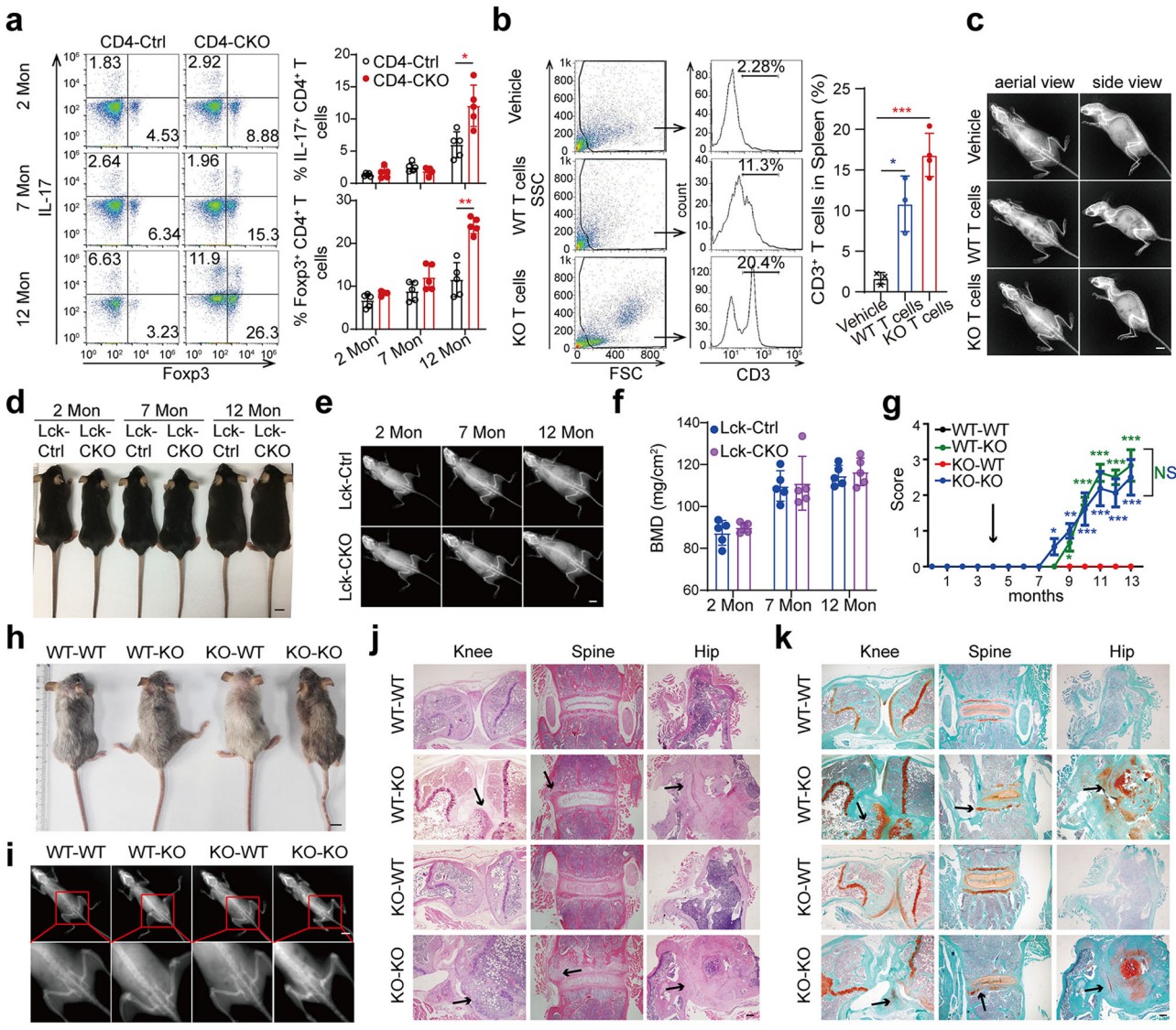

**Fig. 3 Immune system abnormality is not the cause of AS-like bone deformation. a** Flow cytometry analysis of T cell subsets in inguinal lymph nodes. ($n = 5$). **b** Flow cytometry results of CD3$^+$ T percentages in the spleen of 6-month-old recipient NCG mice ($n = 4$). **c** X-ray images of 12-month-old recipient NCG. **d–f** Gross images (**d**), X-ray images (**e**), and Femoral BMD (**f**) of Lck-CKO mice and Lck-Ctrl littermates. ($n = 5$). **g–k** CD4-Ctrl (WT) and CD4-CKO (KO) mice at 4 months were lethally irradiated followed by transferring with bone marrow cells. Irradiated CD4-CKO mice transferred with WT bone marrow cells were labeled as WT-KO, and so on. Pathological scores ($n = 10$) (**g**), Gross images (**h**) and Radiographs (**i**) of 12-month-old chimeras. **c–e**, **h–i** Scale bars: 1 cm. **j, k** H&E and SOFG staining images of chimeras. Scale bars: 200 μm. **j** Arrows show bone deformation. **k** Arrows indicate chondrogenesis and ectopic new bone formation. **a, b, f, g** Data are presented as mean ± SEM. *$p < 0.05$. **$p < 0.01$, ***$p < 0.001$, determined by two-tailed Student's t-test. **c–e**, **h–k** Data are representative of three independent biological replicates.

differentiated from proliferating chondrocytes (Fig. 4a, b). As the bones enlarged further, a secondary ossification center was established in 1-month-old mice, and the GFP$^+$ chondrocytes were still detected in the inner layer of articular cartilage and pre-hypertrophic and hypertrophic zone of the growth plate, as well as in the intervertebral disc of the spine, sacroiliac joint, and crista iliaca (Fig. 4c, d, Supplementary Fig. 11). These GFP$^+$ chondrocytes in CD4-CKO;Rosa26-mTmG mice were SHP2-deficient (Supplementary Fig. 12).

The distribution and proportion of GFP$^+$ chondrocytes were similar in the growth plates of 1-month-old CD4-Cre, 1-month-old CD4-CKO, and 2-month-old CD4-CKO mice, as well as in the spine cartilaginous endplates (Supplementary Fig. 13). This finding suggested that CD4-Cre-mediated SHP2 deficiency in chondrocytes did not disrupt the growth of chondrocytes and skeletal development in young mice. In mature CD4-CKO mice,

there were just ~ 20% chondrocytes were GFP positive in growth plate, as well as in the chondromas (Fig. 4e, f). The bone disease occurred in mature CD4-CKO mice rather than suckling mice with more GFP$^+$ differentiated chondrocytes. These results suggest that CD4-Cre-mediated conditional SHP2 deficiency in chondrocytes is insufficient to induce AS-like bone disease in mice and that other factors, such as maturation or aging, may be involved in the process.

**Premature fusion of the growth plate prevents AS-like bone disease.** Transient inhibition of the Hedgehog pathway by the Smo inhibitor in young mice promoted premature fusion of the epiphyseal plate (growth plate)[38]. In proteoglycan-induced spondylitis (PGIS), a common murine model of AS, inflammation induced pathological new bone formation at spinal enthesis attaching to the

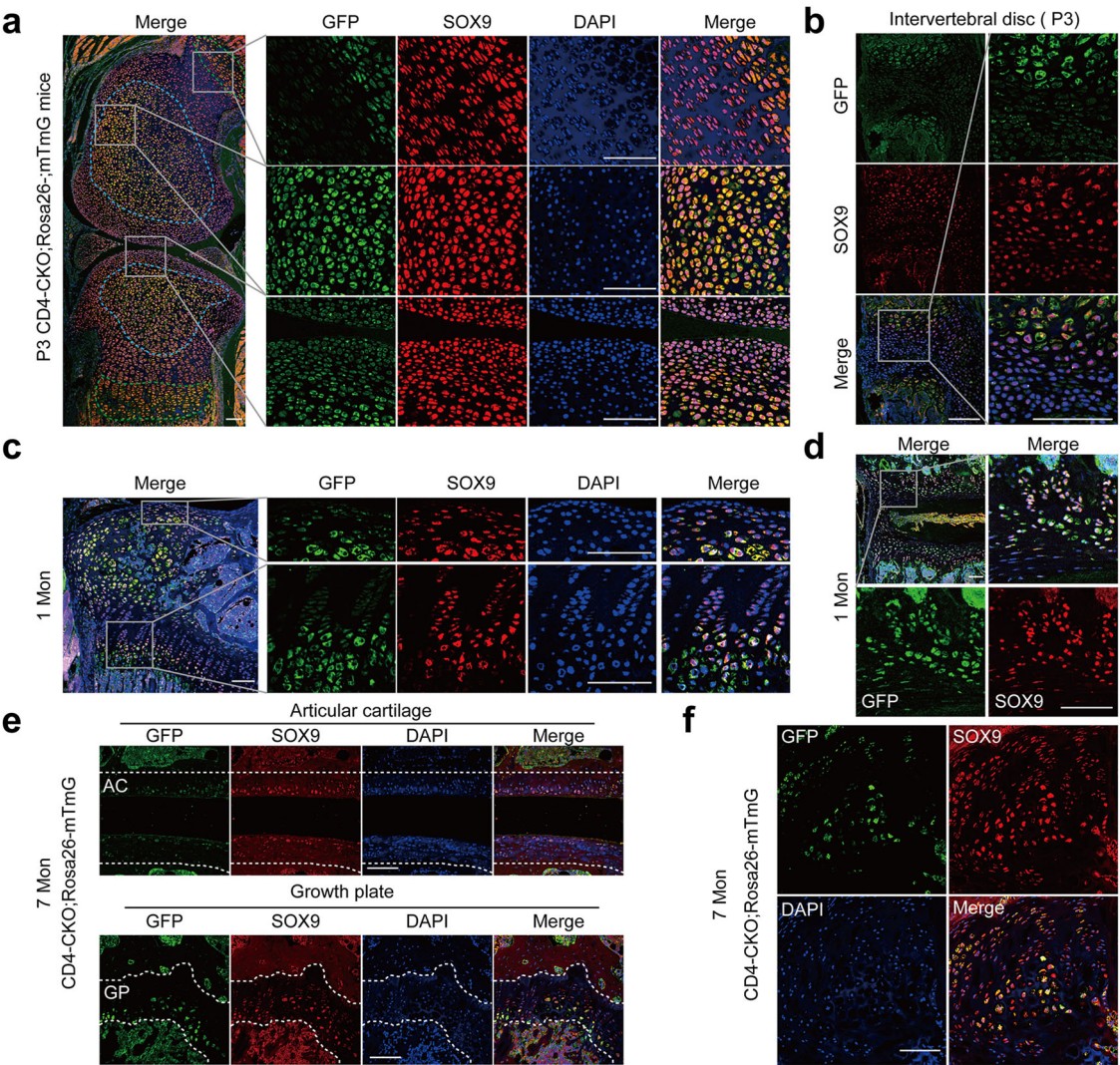

**Fig. 4 CD4-Cre mediates conditional deletion of SHP2 in chondrocytes.** Sagittal section of joints from CD4-CKO; Rosa26-mTmG mice were immunostained with marker protein GFP, transcriptional regulator SOX9 and nuclear counterstain DAPI. **a, b** Immunofluorescence staining images show SOX9 expression in all chondrocytes and GFP expression in hypertrophic chondrocytes in knee and spine section of P3 CD4-CKO; Rosa26-mTmG mice. **a** Blue dotted box shows cartilage mold, and green dotted box indicates hypertrophic zone of the growth plate. **c, d** Representative images of 1-month-old CD4-CKO; Rosa26-mTmG mice show GFP+ chondrocytes consisted in hypertrophic zone of femur and spine. **e, f** Immunofluorescence analysis of articular cartilage and growth plate (**e**) and chondroma (**f**) of the tibia in 7-month-old CD4-CKO; Rosa26-mTmG mice show GFP+ differentiated chondrocytes in the growth plate and articular cartilage of mature CD4-CKO;Rosa26-mTmG mice. **e** The area inside the two white dotted lines is articular cartilage (AC) (upper) and growth plate (GP) (under). **a–f** Scale bars, 100 μm. Data are representative of three independent experiments.

vertebral growth plate through endochondral ossification[39]. To confirm whether growth plate plays a critical role in AS-like bone disease in CD4-CKO mice, we treated 3-week-old mice with Smo inhibitor sonidegib and monitored them for 11 months. Sonidegib treatment caused permanent disruption of bone epiphyses in young mice, including cartilaginous endplates in the spine and growth plates in the femur, tibia, and femoral head (Fig. 5a). Immunofluorescence results showed the absence of SOX9+ chondrocytes in the cartilaginous endplate, as well as in the femoral growth plate, after sonidegib treatment (Fig. 5b, Supplementary Fig. 14). However, it did not influence the structure of the articular cartilage (Supplementary Fig. 14). Eleven months later, sonidegib treatment abolished the AS-like bone disease in CD4-CKO mice (Fig. 5c–f, Supplementary Movie 3). Radiographic and histological analysis showed abolition of structure damage, new bone formation, and osteoporosis in CD4-CKO mice treated with sonidegib (Fig. 5g, h). These data indicated that premature fusion of growth plates prevents the AS-like bone disease in CD4-CKO mice.

**Disorder of the growth plate occurs in CD4-CKO mice and juveniles with enthesitis-related arthritis.** To explore whether CD4-Cre mediated SHP2 deficiency disturbs the growth plate in mice, we injected BrdU solution to label the proliferating cell in vivo. Immunofluorescence results showed more BrdU+ chondrocytes in the growth plate of CD4-CKO;Rosa26-mTmG mice than plate of CD4-Cre;Rosa26-mTmG mice (Fig. 6a, b). In vitro, the primary chondrocytes isolated from wild-type mice were confirmed by SOX9 immunostaining (Fig. 6c). The chondrocytes were transfected with shRNA-Ctrl and shRNA-Ptpn11 lentivirus, respectively, and knockdown of SHP2 was confirmed by western blot (Fig. 6d). SHP2 deficiency promoted the proliferation and gene expression of the Col2a1, Col10a1, Aggrecan, Ihh, and Pthrp in primary mouse chondrocytes and mouse chondrogenic ATDC5 cells (Fig. 6e, f, and Supplementary Fig. 15). Quantitative phase microscopy has shown that mammalian chondrocytes undergo three distinct phases of volume increase during differentiation, and that phase 1 is characterized

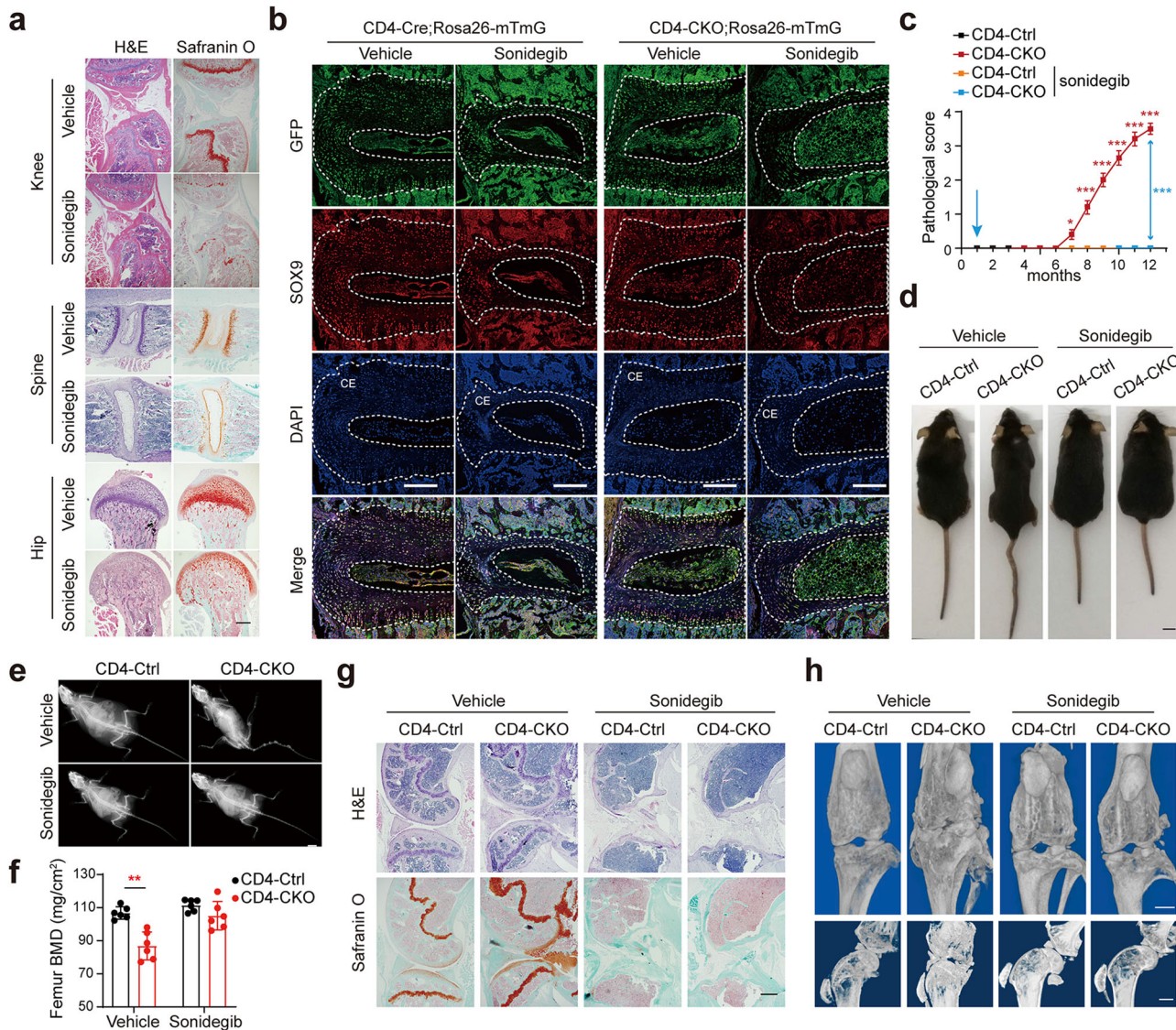

**Fig. 5 Premature fusion of the growth plate prevents AS-like bone disease in CD4-CKO mice.** Three-week-old CD4-CKO;Rosa26-mTmG mice and CD4-Cre;Rosa26-mTmG littermates were orally gavaged every other day with Smo inhibitor, sonidegib (50 mg/kg) for three times. **a** Sagittal joint sections of 2-month-old mice treated with sonidegib or not were stained with H&E and SOFG. Scale bars: 400 μm. **b** Spinal sections of 2-month-old mice were immunostained with marker protein GFP, transcriptional regulator SOX9 and nuclear counterstain DAPI. CE represents cartilaginous endplate between white dotted line. Scale bars: 200 μm. **c** Pathological scores of CD4-CKO mice treated with sonidegib or not ($n = 10$). **d–f** Gross images (**d**), X-ray radiographs (**e**), and femoral BMD ($n = 6$) (**f**) show bone lesions and decreased femoral BMD were merely occurred in CD4-CKO mice. Scale bars: 1 cm. **g** H&E and SOFG staining images. Scale bars: 400 μm. **h** μ-CT radiographs of 12-month-old mice show bone deformation and cartilage abnormalities in CD4-CKO mice without sonidegib treatment and merely smaller skeleton in sonidegib-treated CD4-CKO mice and CD4-Ctrl mice. Scale bars: 1 mm. **c** and **f** Data are presented as mean ± SEM. **c** **\*\*$p < 0.01$, \*\*\*$p < 0.001$, **f** **\*\*$p$-value = 0.0004, determined by two-tailed Student's $t$-test. **a, b, d, e, g, h** Data are representative of three independent biological replicates.

by a threefold enlargement and proportionate increase in dry mass production[40]. To explore which phase of chondrocyte differentiation was regulated by SHP2, we isolated chondrocytes from CD4-CKO;Rosa26-mTmG mice and transfected them with half-dose Cre recombinase adenovirus. Flow cytometry analysis showed the parameters SSC and FSC of GFP+ chondrocytes were larger than GFP− chondrocytes, suggesting that SHP2 deficiency promotes chondrocyte differentiation at phase 1, the first phase of chondrocyte differentiation (Fig. 6g). These results indicate that SHP2 deficiency delays fusion of the growth plate through promoting the proliferation and differentiation of chondrocytes.

Enthesitis-related arthritis (ERA) is a subgroup of juvenile idiopathic arthritis characterized by HLA-B27 positive and enthesitis and accepted as the counterpart of juvenile AS[41,42].

We assessed the epiphyses in patients with ERA and radiological images showed inflammation, bone deformation (white arrows), and thickened growth plates (red arrows) in patients with ERA (Fig. 6h–j, Supplementary Fig. 16). These results suggest that disorder of the growth plate contributes to the occurrence and development of AS.

**Aberrant chondrocytes promote ectopic new bone formation through BMP6/Smad1/5 signaling.** The osteophytes disappeared in CD4-CKO mice with premature growth plate, indicating aberrant chondrocytes promoted pathological new bone formation. Besides Hedgehog proteins, BMPs and Wnt signaling have been shown to be involved in bony fusion of AS[13,43,44].

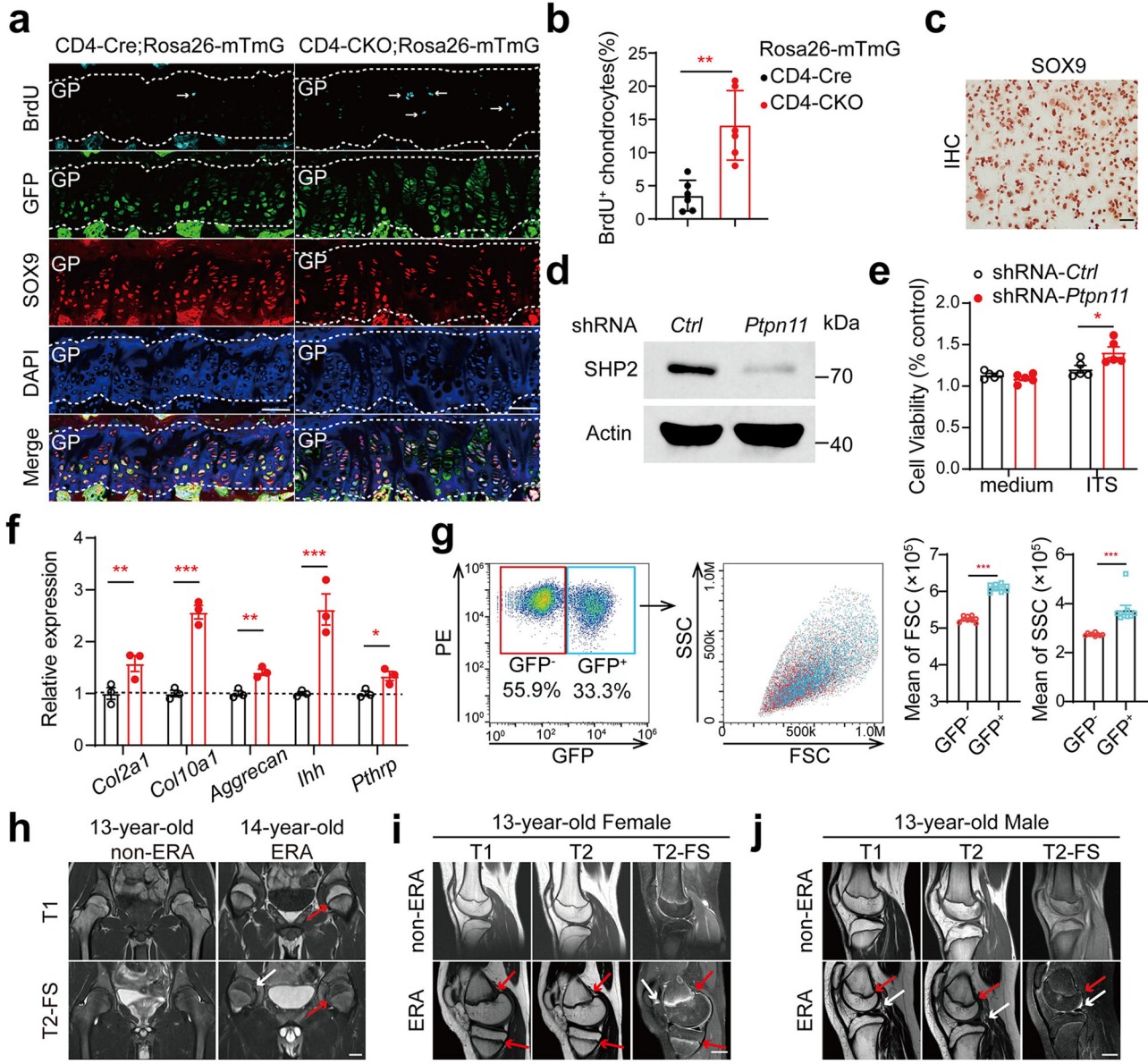

**Fig. 6 Disorder of the growth plate occurs in CD4-CKO mice and juveniles with enthesitis-related arthritis. a, b** Mature mice were injected with 50 mg/kg BrdU intraperitoneally every 8 h for six times. Sections of knee joints were immunostained with BrdU, GFP, SOX9 and DAPI. ($n = 6$). GP: growth plate. The proportions of BrdU$^+$SOX9$^+$ chondrocytes in SOX9$^+$ chondrocytes were measured. **a** Scale bars: 100 μm. **c** Mouse primary chondrocytes of C57BL/6 mice confirmed by SOX9 immunostaining. Scale bars: 100 μm. **d–f** Mouse chondrocytes transfected with shRNA-*Ctrl* or shRNA-*Ptpn11* lentivirus respectively and the SHP2 protein level, cell proliferation activity and the expression of gene *Col2a1*, *Col10a1* and *Aggrecan* were analyzed by Western blot (**d**), MTT activity ($n = 5$) (**e**) and RT-qPCR ($n = 3$) (**f**) respectively. **g** Flow cytometry analysis of Primary chondrocytes isolated from *Ptpn11*$^{f/f}$;Rosa26-mTmG mice were transfected with half-dose Cre recombinase adenovirus. The mean FSC and SSC of GFP$^-$ and GFP$^+$ chondrocytes were calculated. ($n = 9$). **h–j** MRI with T1 weighed, T2 weighed and T2 weighed fat-saturated (T2-FS) sequence of patients with non-ERA or ERA. Hip joints (**h**), and knee joints (**i, j**) images show arthritis, bone deformation (white arrows), and thickener growth plate (red arrows) in joints of patients with ERA than those of age, sex-matched non-ERA patients. Scale bars: 2.5 cm. **b, e, f, g** Data are presented as mean ± SEM. *$p < 0.05$, **$p < 0.01$, ***$p < 0.001$, determined by two-tailed Student's *t*-test. **a, c, d, g–j** Data are representative of three independent biological replicates.

To explore the mechanism of aberrant chondrocyte regulates ectopic new bone formation, we detected the expression of BMP and Wnt proteins expression. RT-qPCR analysis showed significantly higher expression of BMP6 in SHP2-knockdown chondrocytes than in WT chondrocytes (Fig. 7a). It has been reported that SHP2 represses chondrocytic differentiation partly through influencing the phosphorylation and SUMOylation of SOX9 via the PKA signaling pathway[45,46]. In primary mouse chondrocytes, we confirmed that knockdown of SHP2 decreased pERK1/2 level and up-regulated SOX9 abundance (Fig. 7b).

To investigate whether SHP2-deficient chondrocytes promoted osteogenesis, we cocultured mouse bone mesenchymal stem cells (mBMSC) with chondrocytes. The supernatant of SHP2-deficient chondrocytes enhanced phosphorylation of Smad1/5 in mBMSC (Fig. 7c). Alizarin red staining revealed that SHP2-knockdown chondrocytes induced more calcium phosphate—containing nodules formation in mBMSC than WT chondrocytes (Fig. 7d). Alkaline phosphatase (ALP) activity and RT-qPCR analysis showed similar inductive effects on ALP activity and expression of osteogenic gene *Runx2*, *Osx,* and *Ocn*, especially *Runx2*, an

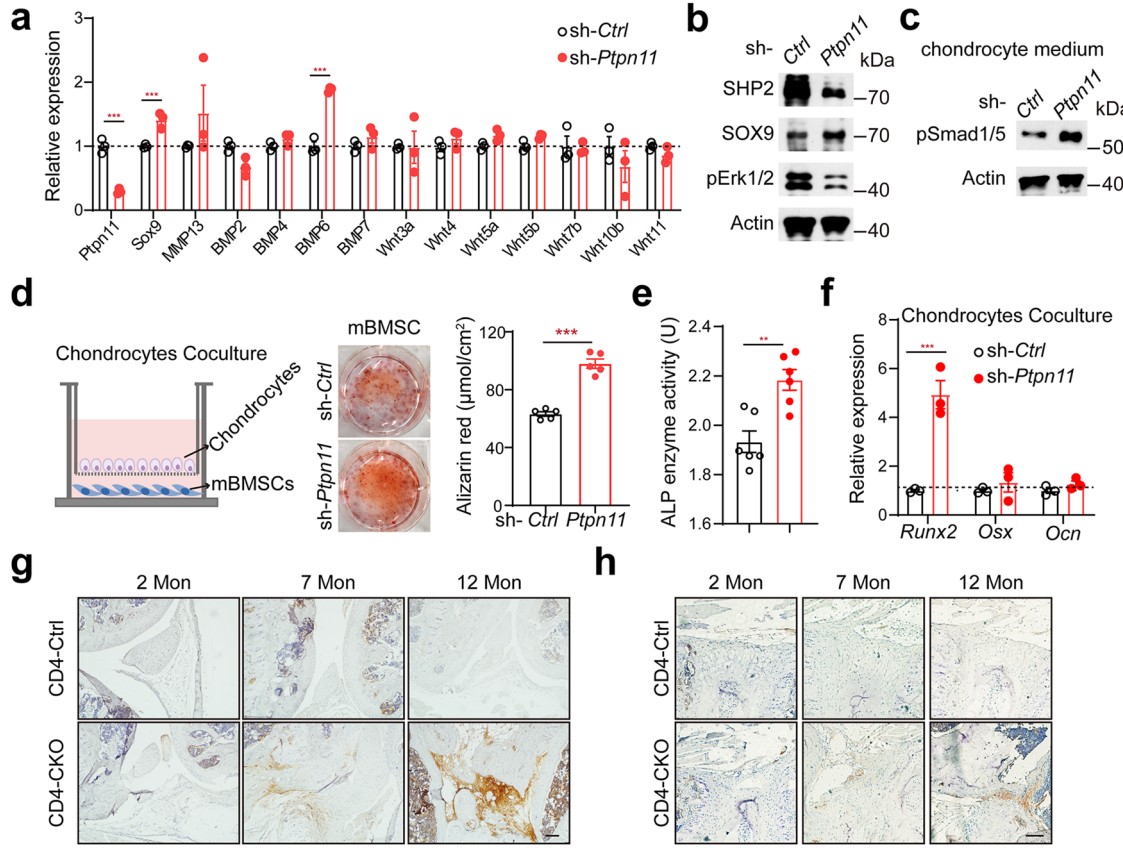

**Fig. 7 Aberrant chondrocytes promote ectopic new bone formation through BMP6/Smad1/5 signaling. a** RT-qPCR analyses of gene expression of chondrocytes after transfected with sh-Ctrl and sh-Ptpn11, respectively. (n = 3). **b** Western blot analyses of chondrocytes of (**a**). **c** western blot analysis of phosphorylation of Smad1/5 protein levels in mBMSC stimulated with chondrocytes supernatant for 30 min. **d** Alizarin Red staining and quantification of mBMSC cocultured with chondrocytes for 14 days. (n = 5). **e, f** ALP activity measurement (n = 6) (**e**) and RT-qPCR analyses (n = 3) (**f**) of osteogenic marker genes in mBMSC cocultured with chondrocytes for 7 days. **e** **p-value = 0.0020. (**g** and **h**) Immunostaining analysis of pSmad1/5 protein levels in knee joint (**g**) and intervertebral disc (**h**). Scale bars: 100 μm. **a**, **d**, **e**, **f** Data are presented as mean ± SEM. *p < 0.05, **p < 0.01, ***p < 0.001, determined by two-tailed Student's t-test. **b**–**d**, **g**, **h** Data are representative of three independent biological replicates.

osteogenic marker gene of naive osteoblasts (Fig. 7e, f). These results indicate SHP2-deficient chondrocytes are more osteoinductive than WT chondrocytes partially through increased BMP6 expression. Immunohistochemistry results showed increased pSmad1/5 level in entheseal tissues and synovial tissue in spine and knee joint of aging CD4-CKO mice (Fig. 7g, h, Supplementary Fig. 17). Meanwhile, Yu et al. reported that chondrogenesis and increased pSmad1/5/8 intensities in the AS interspinous ligaments recently[14]. These results indicate that chondrogenesis mediates AS progression through BMP6/Smad1/5 signaling.

**Targeting chondrocyte retards AS progression in CD4-CKO mice.** Smo inhibitor treatment ameliorated exostoses and meta-chondromatosis in *Ctsk-Cre;Ptpn11^{f/f}* mice[22]. The aberrant chondrocytes promoted ectopic new bone formation in CD4-CKO mice. To investigate whether targeting chondrocytes can retard progression of AS, we treated CD4-CKO mice with Smo inhibitor sonidegib at the initial stage of radiographic progression. Although low-dose sonidegib (50 mg/kg) just slightly improved the bone deformation in CD4-CKO mice (Supplementary Fig. 18), high-dose (100 mg/kg) sonidegib significantly alleviated structure damage in CD4-CKO mice (Fig. 8a, Supplementary Movie 4). Gross and X-ray images revealed that joint stiffness, kyphosis, and scoliosis were improved by high-dose sonidegib treatment (Fig. 8b, c). Meanwhile, femoral BMD measurement indicated that high-dose sonidegib treatment

almost completely abolished bone loss in CD4-CKO mice (Fig. 8d). H&E and SOFG staining showed bony fusion and ectopic new bone formation were attenuated by high-dose sonidegib treatment (Fig. 8e, f, Supplementary Fig. 19a, b). Meanwhile, micro-CT images showed the osteophytes and bony fusion were suppressed by treatment with high-dose sonidegib (Fig. 8g, h, Supplementary Fig. 19c, d). In addition, immuno-histochemical staining demonstrated high pSmad1/5 in joint of CD4-CKO mice were suppressed by high-dose sonidegib treatment (Fig. 8i, Supplementary Fig. 19e). These results suggest blocking endochondral ossification mediated by aberrant chondrocyte retards bony fusion and new bone formation in AS (Fig. 8j).

## Discussion
In this study, we reported ectopic new bone formation and bony fusion, as well as osteoporosis, in mature CD4-CKO mice. These features are the typical features of AS, which could cause spinal ankylosis and even permanent disability. In CD4-CKO mice, SHP2 was knockout in proliferating chondrocytes, resulting in the delay of growth plate fusion ultimately. Aberrant chon-drocytes in growth plate and enthesis mediated pathological new bone formation through endochondral ossification. Furthermore, SHP2-deficient chondrocytes expressed increased BMP6 and promoted ectopic new bone formation through BMP6/pSmad1/5 signaling in vivo and in vitro. Chondrogenesis and increased

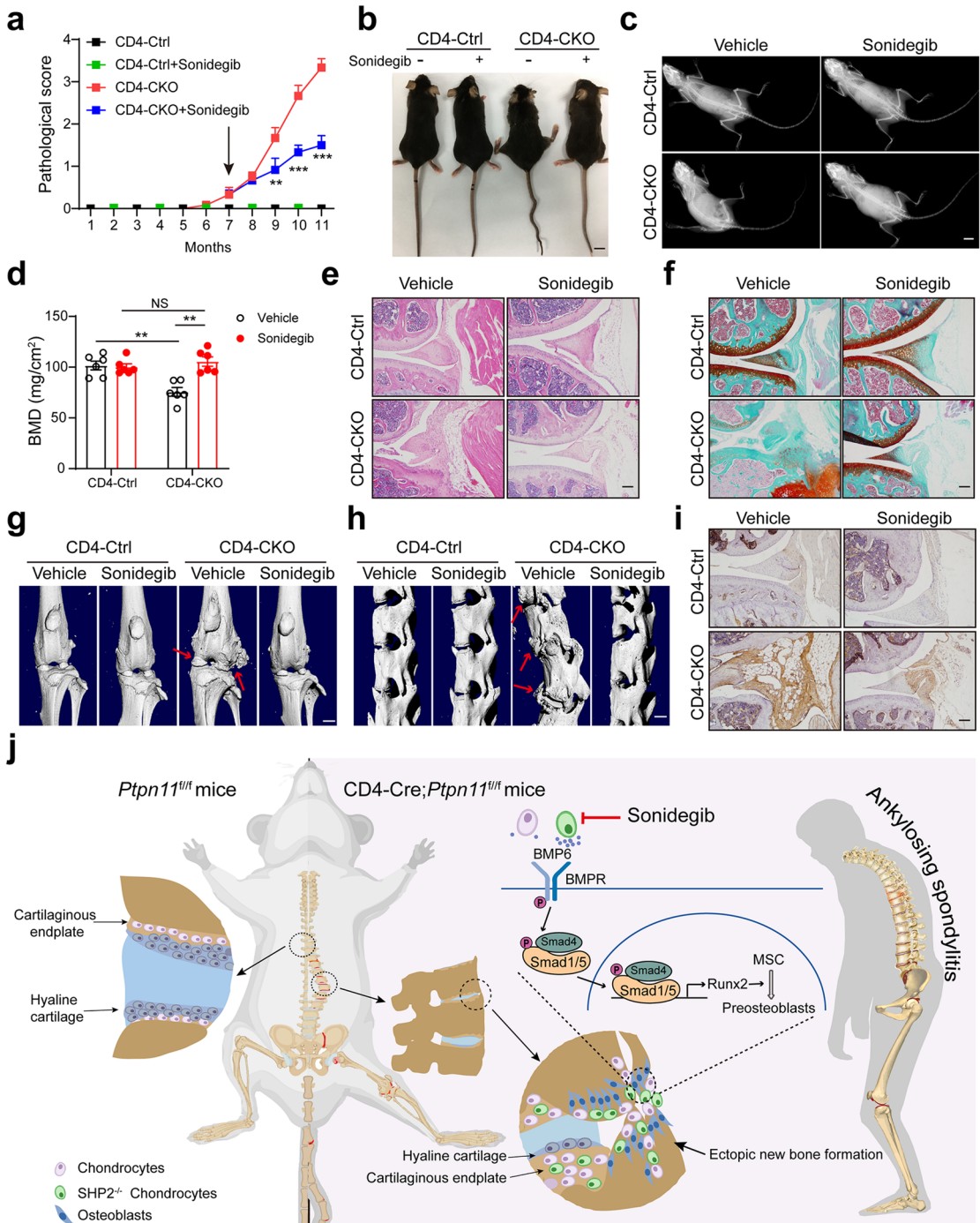

**Fig. 8 Targeting chondrocyte retards AS progression in CD4-CKO mice.** 7-month-old CD4-CKO mice and CD4-Cre littermates were orally gavaged with Smo inhibitor, sonidegib (100 mg/kg) for 4 months. **a** Pathological score ($n = 6$). **b** Gross images. **c** X-ray images. (**b** and **c**) Scale bars: 1 cm. **d** Femur bone mineral density (BMD). ($n = 6$) **e** H&E staining of knee joints. **f** SOFG staining of knee joints. **g** and **h** μ-CT radiographs of knee joint (**g**) and spine (**h**) of 11-month-old mice. Arrows show ectopic new bone formation. Scale bars: 2.5 cm. **i** Immunohistochemical staining of pSmad1/5 protein levels in knee joint of 11-month-old mice. **e**, **f**, **i** Scale bars: 100 μm. **a**, **d** Data are presented as mean ± SEM. **p < 0.01, ***p < 0.001, determined by two-tailed Student's t-test. **b**, **c**, **e**–**i** Data are representative of three independent biological replicates. **j** The mechanism of AS-like bone disease in CD4-CKO mice.

BMP6/pSmad1/5 signaling were also detected in the ligaments of AS[14]. These results indicate the process of pathological new bone formation in CD4-CKO mice was similar to the progress in AS. Targeting chondrocytes through Smo inhibitor significantly alleviated the pathological new bone formation in CD4-CKO mice. These findings suggest that targeting chondrocytes may be an effective therapeutic strategy for retarding pathological new bone formation and radiographic progression in AS.

It has been reported that SHP2 deletion in chondrocytes led to metachondromatosis and scoliosis in mice within 1 month[23,25]. Functionally, SHP2 depletion delayed the terminal differentiation of chondrocytes[24]. Mechanistically, SHP2 deletion in chondrocytes led to increased expression of *Ihh* and *Pthrp*, along with decreased activity of p-Erk1/2 signaling[22,24]. SHP2 regulates SOX9 abundance mediated at least in part via the PKA signaling pathway in the process of chondrocytes differentiated from

osteochondroprogenitors[26]. In addition, CD4-Cre-mediated conditional Sos1/2 or Erk1/2 deletion in chondrocytes resulted in cartilage tumors, which was similar to the cartilage tumors in wrist bones of 6-month-old CD4-CKO mice[34–37]. These results established that SHP2-mediated Ras/Erk pathway signaling is critical for the regulation of differentiation of chondrocytes and skeletal development.

Although the T cell-independent cartilage tumor has been reported in CD4-CKO mice[37], the source of the cartilage tumor and the subsequent ectopic new bone formation have not been mentioned. We found CD4-expressing proliferating chondrocytes through immunostaining of Cre recombinase. Massive SHP2-deficient pre-hypertrophic and hypertrophic chondrocytes differentiated from CD4-expressing proliferating chondrocytes and disturbed the fusion process of growth plate. The chondromas were abolished by premature growth plate, suggesting ectopic chondrogenic differentiation was facilitated by activated chondrocytes in the epiphyseal plate in aged mice. Subsequently, endochondral ossification mediated by chondrocytes promoted bone structure damage and ectopic new bone formation, resulting in bony fusion and ankyloses.

Animal models including proteoglycan aggrecan-induced arthritis (PGIA), HLA-B27 transgene rats, TNF-α transgenic mice, and DBA/1 mice, have been designed to investigate the pathogenesis of AS[43,47–49]. However, the phenotypes of these models resemble those of arthritis more than AS-like bone fusion. Long-term immunization with proteoglycan aggrecan has been shown to induce partial spondylitis (PGISp) in BALB/c mice with phenotypes that develop independently of arthritis and recapitulate those in human AS patients[39]. However, this model suffers from several drawbacks, including multiple injections, low success rate, and difficulty to obtain proteoglycan aggrecan. Although inflammation is an indispensable feature of AS and syndesmophytes are likely to occur at sites with active inflammation, the ability of current anti-inflammation treatment of arresting the radiological progression of AS is debatable. To explore new treatment of AS after the detection of inflammation in CD4-CKO mice, we attempted to use Iguratimod, an anti-inflammatory drug for rheumatoid arthritis[31], to retard the bone disease in CD4-CKO mice. As a result, there was no improvement upon Iguratimod treatment. Moreover, it has been reported that the majority of new vertebral syndesmophytes develop at sites with no evidence of previous inflammation[11,12], suggesting new bone formation in AS is not merely induced by inflammation. Therefore, CD4-CKO mice would be a platform to screen the potential targets that directly arrest the bony fusion in AS.

Blocking the osteogenesis process would be the direct mode to arrest the radiological progression of AS. Chondrogenesis mediates pathological new bone formation in ligaments of AS, as well as in CD4-CKO mice. The hedgehog signaling pathway regulates chondrogenesis and endochondral ossification through coordinating proliferation and differentiation of chondrocytes[50]. Smoothened (Smo) is a transmembrane protein that transduces all Hh signals. Smo inhibitor sonidegib alleviated the metachondromatosis induced by SHP2 deletion in chondroid progenitors. In CD4-CKO mice, sonidegib treatment significantly alleviated or even abolished bony fusion. These results suggest targeting chondrogenesis is efficient to arrest the radiographic progression of AS.

Two paradoxes have been reported in the clinical features of AS: (1) Cartilage degeneration is accompanied by osteoproliferation; and (2) Osteoporosis juxtaposes with bone formation in AS[51–53]. The mature and aged CD4-CKO mice showed cartilage degeneration of the articular cartilage and thickened growth plate in the knee sections, indicating that degeneration is limited to articular cartilage and osteoproliferation is mediated by growth plate. Induction of premature fusion of the growth plate with Smo inhibitor in CD4-CKO mice eliminated cartilage degeneration and bone deformation, suggesting that cartilage degeneration is probably a complication of bone deformation. Increases in osteoclast numbers in CD4-CKO mice lead to osteoporosis and this phenotype is abolished by premature fusion of the growth plate, suggesting that osteoproliferation in joints facilitates osteoclast differentiation through crosstalk between osteoblasts and osteoclasts and/or between chondrocytes and osteoclasts.

Taken together, these findings indicate that bone disease in aging CD4-CKO mice closely mimics AS. To the best of our knowledge, this is the first reported spontaneous animal model of AS which can become a valuable platform for the research of pathological new bone formation in AS. Mechanistically, CD4-Cre mediated SHP2 deficiency in proliferating chondrocytes, which differentiated into pre-hypertrophic and hypertrophic chondrocytes, resulting in delay of growth plate fusion. The active chondrogenesis in growth plate and enthesis induced osteoproliferation and ectopic new bone formation partly through BMP6/Smad1/5 signaling. The osteophytes bridged the joint cavity, resulting in joint stiffness and even permanent disability. Smo inhibitor sonidegib abolished the uncoordinated chondrogenesis and significantly alleviated the pathological new bone formation in CD4-CKO mice. These findings suggest that targeting disordered chondrocytes is a promising avenue for arresting the radiological progression of AS. Our data collectively indicate that blockade of chondrogenesis by sonidegib would be a drug repurposing strategy for AS treatment.

## Methods

**Mice**. CD4-CKO mice were generated by crossing *Ptpn11*^f/f mice with CD4-Cre transgenic mice SHP2 conditional deletion in T cells were confirmed previously[54]. Lck-Cre;*Ptpn11*^f/f mice were generated by crossing *Ptpn11*^f/f mice with Lck-Cre transgenic mice. CD4-CKO;Rosa26-mTmG mice were generated by crossing CD4-CKO mice with Rosa26-mTmG transgenic mice. These mice were all in C57BL/6 background and purchased from GemPharmatech Co. Ltd. (Nanjing, China). The animals were maintained with free access to pellet food and water in plastic cages at 21 ± 2 °C and kept on a 12 h light–dark cycle. All mice are harbored in the specific pathogen-free facility in Nanjing University. The animal use and the experimental protocols were reviewed and approved by the Animal Care Committee of the Nanjing University in accordance with the Institutional Animal Care and Use Committee guidelines. Animal welfare and experimental procedures were carried out in accordance with the Guide for the Care and Use of Laboratory Animals (National Institutes of Health, USA) and the related ethical regulations of our university. All efforts were made to reduce the number of animals used and to minimize animal suffering.

**Human specimens**. The study and protocols were carried out in accordance with the ethical guidelines of the 1975 Declaration of Helsinki Principles and were approved by the Ethics Institutional Review Board of Affiliated Hospital of Nanjing University of Chinese Medicine (study number 2018NL-106-02) and Children's Hospital of Nanjing Medical University (study number 202008041-1). Written informed consents were obtained from all patients. Magnetic resonance imaging (MRI) of patients with AS were from three patients (male, 23–52 years old) diagnosed with AS according to Bath Ankylosing Spondylitis Disease Activity Index (BASDAI). Patients with ERA and non-ERA (13–14 years old) were patients of the Department of Rheumatology and Immunology with sacroiliac arthritis or not respectively. The images were collected by 1.5 T MRI scanner (Siemens).

**Scoring severity of bone disease**. Scored on a scale from 0 to 5: 0, none; 1, enlarged pelvic incidence angle; 2, slight stiffness of knee joints; 3, scoliosis and kyphosis and moderate stiffness of knee joints; 4, marked scoliosis and kyphosis, ankylosis of knee and hip joints; 5, ankylosis of lower part of the body and spine.

**Radiography analysis**. Mice were scanned using faxitron X-ray system (IVIS Lumina XR), and BMD of femurs were measured by dual-energy X-ray absorptiometry system. Undecalcified specimens were scanned by X-ray microtomography (Skyscan 1176, Belgium) with the following settings: 50 kV, 0.5 mm aluminum filter, and 9 μm isotopic resolution. Trabecular bone parameters of the femur were calculated using CTan software (Bruker microCT). Three-dimensional reconstruction images were produced with CTvol (Bruker microCT).

**Skeletal preparations**. Mice skeletons were fixed in 99% ethanol and immersed in acetone, and then incubated in staining buffer for 3 days. The staining buffer was mixed with 5 mL 0.3% Alcian blue 8GX (Sigma-Aldrich, in 70% ethanol), 5 mL 0.1% Alcian red S (Sigma-Aldrich, in 95% ethanol), 5 mL glacial acetic acid, and 85 mL 70% ethanol.

**Histological and TRAP staining**. For histological analysis, the bone was fixed with 10% buffered formalin and decalcified by 10% EDTA, dehydrated in gradient ethanol, embedded with paraffin and sliced, and then used for H&E or SOFG staining. TRAP staining was performed using the Acid Phosphatase (TRAP) Kit according to the manufacturer's instructions (Sigma-Aldrich).

**Immunohistochemistry and Immunofluorescence staining**. Paraffin-embedded slices (5 μm) were de-waxed, hydrated and then the antigen was retrieved using sodium citrate. After blocked with 5% goat serum, the slices were incubated with the following antibodies: anti-osteocalcin (Santa Cruz Biotechnology, sc-376835), anti-SOX9 (Abcam, ab76997), anti-GFP (Cell Signaling Technology, 2555), anti-BrdU (Biolegend, 364102), anti-Cre (Cell Signaling Technology, 15036), anti-SHP2 (Santa Cruz Biotechnology, sc-7384), anti-pSmad1/5 (Cell Signaling Technology, 9516) antibody at 4 °C overnight. Then, immunofluorescence staining slices were probed with secondary antibodies: goat anti-mouse IgG2a conjugated to Alexa Fluor 594 (Invitrogen, A-21135), goat anti-mouse IgG1 conjugated to Alexa Fluor 647 (Invitrogen, A-21240), goat anti-rabbit IgG conjugated to Alexa Fluor 488 (Invitrogen, A-11008). After counterstained with DAPI, the slices were imaged with a confocal laser scanning microscope (Zeiss LSM880 with Airyscan). Immunohistochemistry staining slices were incubated with HRP-conjugated secondary antibodies after primary antibody, followed by DAB substrate solution and Hematoxylin. The images were obtained with microscopy.

**Enzyme-linked immunosorbent assay (ELISA)**. The blood was centrifuged at 1000 g for 15 min and the serum was collected. The cytokine levels in serum were determined using an ELISA kit according to the recommendations of the manufacture.

**Flow cytometry analysis and antibodies**. The mononuclear cells were cultured in RPMI 1640 medium supplemented with 10% FBS and Cell Stimulation Cocktail (plus protein transport inhibitors) (eBioscience) for 5 h at 37 °C in a thermal incubator under 5% $CO_2$. The cells were stained with cell surface marker, including CD3 (17A2), CD4 (GK1.5), and CD8 (53–6.7). For intracellular staining, fixing and permeabilization were performed using Intracellular Fixation and Permeabilization Buffer Set according to the manufacturer's instructions (eBioscience). The cells were then incubated with anti-FcγRIII/II antibody (2.4G2, Fc block) followed by fluorochrome-conjugated antibodies, comprising IFN-γ (XMG1.2), IL-2 (JES6-5H4), IL-17 (TC11-18H10.1), and Foxp3 (MF-14). Bone marrow cells were labeled with antibodies to lineage markers, including CD45 (30-F11), CD11b (M1/70), CD3 (17A2), CD4 (GK1.5), B220 (RA3-6B2), CD19 (6D5), NK1.1 (PK136), Ly6G (1A8), CD31 (MEC13.3), CD11c (N418). Fluorochrome-conjugated antibodies were purchased from BioLegend, eBioscience, or BD Pharmingen. Mouse primary chondrocytes were digested from knee joints and ribs of 4-day-old CD4-CKO; Rosa26-mTmG mice and adherent cells were analyzed by flow cytometry. Data were acquired by either a FACSaria II (BD) or Attune NxT (Life Technologies) flow cytometer and analyzed by FACSDiva or FlowJo (BD) software.

**T cell transfer model**. CD3$^+$ T cells were isolated from spleen cells of 4-month-old CD4-Ctrl and CD4-CKO mice by positive magnetic separation according to the manufacturer's instructions (Miltenyi Biotec). Severe immunodeficient NCG (NOD-Prkdc$^{em26Cd52}$Il2rg$^{em26Cd22}$/Nju) mice were kindly provided by Gem-Pharmatech Co. Ltd. (Nanjing, China). Age, sex-matched NCG mice were intravenously transferred with purified CD3$^+$ T cells (5 × 10$^6$ cells per mouse) from CD4-Ctrl and CD4-CKO mice, respectively. The recipient mice were fed and monitored for 8 months after transfection.

**Bone marrow transplant**. Four-month-old CD4-Ctrl and CD4-CKO mice were used as recipients or donors and bone marrow transfer was performed as described[55]. Briefly, bone marrow cells (1 × 10$^7$ cells per mouse) isolated from CD4-Ctrl mice or CD4-CKO mice were intravenously transferred into lethally irradiated CD4-Ctrl mice and CD4-CKO mice (8 Gy), respectively. The chimeric mice were fed with drinking water supplemented with antibiotics and 5% sucrose for 2 weeks to improve bone marrow transplantation outcomes. The chimeric mice were then monitored for 8 months after transplantation.

**Primary murine chondrocytes isolation and cell culture**. The primary mouse chondrocytes were isolated from knee joints and ribs of 4-day-old mice[56,57]. Principally, the knee joints and ribs were dislocated and eliminated all the soft tissues. The cartilage was incubated with 2 mg/mL pronase for 45 min followed by 3 mg/mL collagenase D for 1 h at 37 °C to detach soft tissues. Retrieved cartilage pieces were washed with PBS buffer, then placed in fresh 3 mg/mL collagenase D

digestion solution at 37 °C in a thermal incubator under 5% $CO_2$ for 4 h. The chondrocytes were resuspended in DMEM medium supplemented with 2 mM L-Gln, 50 U/mL penicillin, 0.05 mg/mL streptomycin and 10% FBS (Fetal bovine serum) and seeded in a culture dish.

**Mouse chondrogenic cell line**. ATDC5 cells were cultured in DMEM/F12 medium containing 10% FBS in the presence of insulin, transferrin, and sodium selenite supplement (Sigma-Aldrich) for 21 days to induce hypertrophic differentiation as described previously[58,59].

**Mouse bone mesenchymal stem cells (mBMSC)**. Mouse bone marrow was isolated from femur and tibia and plated in a 6-well-plate to induce osteogenesis. The cells were cultured in an osteogenic medium consisting of a DMEM medium supplemented with 10% FBS, 50 μM L-ascorbic acid, 10 nM dexamethasone, and 10 mM β-glycerophosphate. The medium was changed every other day. For quantification, the calcium deposits were destained with 10% cetylpyridinium chloride in 10 mM sodium phosphate (pH 7.0), and the absorbance of the samples was measured at 450 nm.

**Lentiviral transduction**. The lentivirus for shRNA-*Ptpn11* and shRNA-scramble (shRNA-*Ctrl*) were purchased from Shanghai Obio Technology Co. Ltd. (Shanghai, China). The hairpin sequence targeting the *Ptpn11* gene is CCTGATGAG-TATGCGCTCAAA. Lentiviral transduction was performed as described previously[21]. Briefly, mouse primary chondrocytes and ATDC5 cells were transfected with shRNA-*Ctrl* or shRNA-*Ptpn11* lentiviruses. The cells were infected with lentivirus for 72 h and selected with puromycin (Sigma-Aldrich, 10 μg/mL) for 14 days. Then, the protein level and gene expression were detected by immunoblotting and qPCR, respectively.

**Immunoblotting**. Immunoblot assay was performed as described previously[60]. Briefly, proteins extracted in lysis buffer were separated by SDS–polyacrylamide gel electrophoresis and electrophoretically transferred onto polyvinylidene difluoride membranes. The membranes were probed with antibodies overnight at 4 °C, and then incubated with a horseradish peroxidase-coupled secondary antibody. Detection was performed using a LumiGLO chemiluminescent substrate system. The following antibodies were used at indicated dilutions: anti-SHP2 (Santa Cruz Biotechnology, sc-7384), anti-SOX9 (Abcam, ab76997), anti-pSmad1/5 (Cell Signaling Technology, 9516), anti-p-Erk1/2 (Cell Signaling Technology, 4370), anti-ACTIN (Abmart, M20010).

**Reverse transcriptase-PCR and quantitative PCR**. Total RNA was extracted from the cells using Tripure reagent (Roche Diagnostics, Indianapolis, IN) as described by the manufacturer. Single-stranded cDNA was synthesized total RNA by reverse transcription. PCR cycling conditions: 94 °C for 3 min, 35 cycles of 94 °C for 30 s, 62 °C for 40 s, and 72 °C for 1 min. The expression of β-actin was used to normalize the data. Primer sequences are shown in Supplementary Table 1.

**Adenovirus transfection**. The adenovirus expresses both Cre recombinase were purchased from Shanghai Obio Technology Co. Ltd. (Shanghai, China). Mouse primary chondrocytes were infected with adenovirus at a multiplicity of infection of 40 and analyzed by flow cytometry 3 days later.

**Statistics**. Statistical analyses were performed on Prism (GraphPad Software) v8.2.1. Statistical significance was analyzed using a two-tailed Student's *t*-test, and $P < 0.05$ was considered to be statistically significant.

**Reporting summary**. Further information on research design is available in the Nature Research Reporting Summary linked to this article.

## Data availability

All data supporting this study described in this manuscript are available in the article and in the Supplementary Information/Source data file. Source data are provided with this paper.

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

## Acknowledgements
We thank Prof. Zichun Hua (Nanjing University, Nanjing, China) for presenting CD4-Cre transgenic mice. We also thank Dr. Jian Luo (East China Normal University, Shanghai, China) for presenting ATDC5 cells. This work was supported by the National Natural Science Foundation of China (Nos. 81673436, 91853109, 81872877, 81730100, 81704099), and the Mountain-Climbing Talents Project of Nanjing University.

## Author contributions
Y.S., Q.X. and H.Y. conceived and supervised the study. F.S., Q.L., Y.Z. and Z.F. performed the cell line experiments and animal experiments and analyzed the data. W.C., S.L., X.L., W.G. and G.S.F. provided the experimental materials. F. S. and Y. S. wrote the manuscript. All authors discussed the results and commented on the manuscript.

## Competing interests
Y.S. and Q.X. have a patent pending on use of sonidegib in ankylosing spondylitis. The other authors declare no competing interests.
