## [Peer Review File · Nature Communications]

Reviewers' Comments:

Reviewer #1:

Remarks to the Author:

The manuscript by Shao and colleagues presents a novel animal model for ankylosing spondylitis and provide also a novel therapeutic approach to treat bone fusion.

-In the overall the manuscript is not well written, and the meaning of several sentences is not clear. E.g.: A novel population of CD4-Cre-expressing hypertrophic chondrocytes was SHP2 deficiency and delayed fusion of growth plate. Premature growth plate fusion by Smo inhibitor sonidegib blocked the AS-like bone disease. The CD4 glycoprotein is expressed as a T cell receptor on the cell membrane. page 8 the paragraph relative to chimera description is quite confusing.

The manuscript needs a revision from a native English speaker.

-The presented CKO model as an animal model for AS is not convincing and this fact limits its use. Although the morphological and histological aberrations of the skeleton and joints observed closely recall what observed in AS in human, a more effective comparison with human samples should be provided. Moreover, comparative analysis with human AS samples (or data from literature) at the cellular and molecular level is totally missing, and this may lead to misinterpretations. Whereas the knowledge of the molecular basis of AS is poorly understood yet, autoimmunity and inflammation are two well-known hallmarks of such pathology that should be taken into account in the development of an animal model to be used in the development of effective targeted therapies. For example, pro-inflammatory profile of CKO model should be provided. Such data should also support the use of Igaratimod (pag. 7).

The level of novelty is not very high. One relevant research paper on the role of SHP2 in chondrocytes is missing (<https://doi.org/10.1371/journal.pgen.1004364>). Major findings from the study by Bowen and colleagues are delay in the terminal differentiation of chondrocytes, abnormal organization of chondrocyte maturation zone; significantly increased Hh signalling.

Page 9..please clarify the sentence "...SHP2 deficiency hardly disrupt....", Is SHP2 conditionally knocked-out only in hyperthrophic chondrocyte?

Page 11 Which are the basis of the treatment with Hh inhibitor Sonidegib?

Materials and methods descriptions as well as statistical analysis are adequate

Reviewer #2:

Remarks to the Author:

In this manuscript Dr. Shao and colleagues provide a characterization of CD4-Cre;Ptpn11f/f mice as a model of ankylosing spondylitis. The investigators noted that the mice developed spontaneous kyphosis, abnormal gait, and bony fusion. They exam lineage, hematopoietic cell dependency, inhibitors and metabolic profiles. The inclusion of human data for correlation is an additional strength of the manuscript.

In general the studies are well done. There are several elegant studies including lineage tracing indicating that the pathology is most likely initiated by chondrocytes and the inclusion of Ick-cre control experiments is much appreciated as a control for the effects from a T cell population.

The manuscript lacks clarity on a few issues.

Although the authors may not have anticipated the phenotype in retrospect the literature supports their findings: I) Loss of PTPN11 (SHP2) in mice or in patients with PTPN11 mutations (metachondromatosis) has been reported to cause cartilage growths/benign tumors on the bone surface (exostoses) and enchondromas. II) CD4 cre has been described to drive chondrocyte deletion of SOS and ERK (Front. Immunol. 8:343; Front. Immunol. 8:482), which is acknowledged in the results and should be in the introduction or discussion. III) Both SHP2 and ERK have been described as regulating chondrocyte terminal differentiation, growth plate architecture and skeletal cell fates (summarized in PLoS Genet. 2014 May; 10(5): e1004364). This is also buried in the manuscript but the manuscript would be easier to understand if these points were in the introduction. IV) SHP2 deficient chondrocytes can direct endochondromas in trans with other sufficient chondrocytes, so only some of the cartilage needs to be CKD for pathology to occur (PLoS Genet. 2014 May; 10(5): e1004364). V) SHP2 deficiency in fibroblasts lining the bone can cause similar pathology.

Outlining the prior work does not detract from the current extensive analyses and is currently scattered throughout the manuscript making the logic fragmented and difficult to follow.

The list of AS mouse models the authors should also list the transmembrane TNF models.

The CD4-CKO mouse at 12 months with splayed feet demonstrates the morbidity with this pathology. Was there premature mortality or paresis associated with the spinal pathology? How frequently did this occur?

There appears to be bone erosion in the histology which is part of the pathology in human AS and should be specifically mentioned with the TRAP staining.

There should be some measures of serum inflammatory cytokines (minimum TNF, KC, IL-6, IP10, MCP-1), and recommend a minimal serologic panel of RF and ANA. The serologic markers are demonstration of classic "seronegative" axial arthritis. This is for characterization of the model in relation to the human condition.

The inhibition of pathology with a smoothed antagonist speaks further to a cartilage associated pathology as IHH has been implicated in cartilage tumors and aberrant growth. However the partial effect of the smoothed inhibitor does not warrant the conclusion that "chondroma plays a trivial role in AS-like bone lesions in CD4-CKO mice." as this is likely a complex process, the treatment at 2 months was effective but less so at 6 months which allowed time for additional processes, and the CD4-CKO chondrocytes are likely acting in

trans on the other chondrocytes. In addition the scores in Supplemental figure 12 for the control group are relatively low compared to the other experiments and the difference would have appeared to be greater. The conclusion for treatment of the younger group: "Taken together, the data indicated that induction of premature fusion of growth plates prevent the AS-like bone disease in CD4-CKO mice" is not precise as the data reflect that the premature fusion of the growth plate is a marker of the mechanisms associated with the formation of chondromas, and abnormal bone formation and fusion. Fusion of the growth plate in itself is unlikely to arrest all of the pathologies.

The choice of Iguratimod as a treatment agent should be discussed as this is not in the treatment regimens for AS (as outlined by the authors). This treatment abrogated weight loss and may have had an effect on TNF in this model. The other treatments had BMD checked. Was it also checked for this treatment group?

Please review the manuscript for typographical errors and a few are listed:

Osteoallosis, osteophyte accumulation, and osteoporosis in CD4-CKO mice but not in Sonidegib-treated CD4-CKO mice (Figure 5G and H)-this sentence is missing a verb.

The metabolites were significant differences- do the authors mean there were significant differences between metabolites or the metabolites themselves were different?

Figure 8 panel B uric should be uric

Supplemental figure 15

Suramroin change to Suramrin

Reviewer #3:

Remarks to the Author:

In the present work, the authors reported that deletion of SHP2 in CD4+ cells caused age-related AS-like bone disease which had similar pathology to AS patients. This phenotype is dependent of Shp2 deficiency in hypertrophic chondrocytes but not in CD4+ T cells. Mechanistically, SHP2-deficiency led to the increased purine metabolism in chondrocytes, thus promoting bone deformation through paracrine mechanisms. In general, the findings in this study are novel, and may provide a potential strategy for arresting new bone formation and radiological progression of AS. However, there are still some concerns need to be addressed to strengthen the present work.

1.The authors came to the conclusion that SHP2-deficient T cells were unlikely the effector cells that induce the AS-like bone lesions in CD4-CKO mice by adoptive transfer experiment. However, it is still possible that Shp2 deletion in CD4+ T cells (or other CD4+ immune cells, such as a subset of NK cells) caused immune abnormality in CD4-CKO mice, which indirectly affected the function of Shp2-deficient chondrocytes but not Shp2-sufficient chondrocytes. In other words, both CD4+ T cell- and chondrocyte-expressed Shp2 are responsible for the phenotype observed in this work. It is best to design more experiments to exclude this possibility.

2. In the pathogenesis of AS, which cells are the main source of purine or purine metabolites?

3. The authors observed that SHP2-knockdown in chondrocytes resulted in the increased production of cyclic AMP, hypoxanthine, and xanthine. What is the possible underlying mechanism(s)?

4. In the final section of the Results, the authors only found an increased purine metabolites in the serum of patients with AS. It is insufficient to come to the conclusion that purine metabolism facilitates the radiological progression of AS through paracrine signaling. The enhanced level of purine metabolites might be secondary to the pathogenesis of AS, not the cause. The authors should provide more evidence (or at least cite some previous papers) to support their conclusion.

5. It has been reported that Shp2 deficient in CD4+ Cells causes cartilage malfunction without affecting T Cell development and functions. ("Ptpn11 Deletion in CD4+ Cells Does Not Affect T Cell Development and Functions but Causes Cartilage Tumors in a T Cell-Independent Manner. Front. Immunol.2017"). The authors should put more discussion on this and describe how this study relates to your present work.

Reviewer #4:

Remarks to the Author:

The comments provided for this peer-review is heavily focused on the reporting of non-targeted metabolomics performed alongside a number of immunological and gross phenotypic parameters. Notably, non-targeted metabolomics was utilized to support the findings for purine metabolites as the responsible metabolic pathway to facilitate AS-like bone disease in CD4-CKO mice AND that was differentiating AS patients from healthy controls. A number of major additional points of clarification are needed to support the approach and the results interpretations prior to acceptance for publication.

1. The rationale and justification for limiting the non-targeted metabolomics approach to the targeted analysis of purine nucleotides is needed. There are likely to be a number of other nucleotide as well as other metabolic pathways classes involved in distinguishing this phenotype that are not mentioned. Figure 8 was incomplete as the multiple other metabolic pathways identified that distinguish human patients from healthy controls were not included for enhanced understanding of the relative importance of selected metabolites.

2. What are the total number of metabolites that were significantly different for both animal and human datasets (across all metabolic pathways examined using metabolomics)? The data as shown does not support that the differential metabolic productions (DMPs) were primarily important to purine metabolism.

3. The methods section states use of both plasma and serum, and this should be clearly indicated which was applied for the data presented in mice and human. Additionally, if it is

plasma, then the anticoagulant used during blood collection should also be mentioned for reproducibility.

4. There are no method details provided regarding the process for metabolite identification (assuming library database is available for curation?). Were commercial standards used for library, what is the level of confidence for confirmation of the 43 metabolite identifications that varied significantly in the serum between the mature CD4-CKO mice and CD4-Ctrl littermates? Was the same method for identification used for the human sera?

5. How were the raw non-targeted metabolomics data analyzed, was this prior to metabolite identification, were the fold differences on relative abundances normalized or median scaled? What does the scale represent for Figure 7D? Assuming all statistics were applied on fold differences between groups or were relative abundances used in some cases?

6. What are the group comparisons for pathway enrichments shown in Figure 7C and 7F, and what are the total number of metabolites included in these pathway analyses. The analysis and results should include the relative importance of the purine metabolic pathway enrichment to other scores for nucleotide or metabolic pathways. Inclusion of the additional metabolic pathways affected will bolster the findings for a role in purine metabolism as currently this appears to be a targeted analysis using a non-targeted metabolomics dataset.

7. The description of the unsupervised cluster and pathway analysis was incomplete. Additionally, no PCA or OPLS-DA was provided for animals as was included for humans. Was there a separation or heat-map completed for more pathways?

8. What statistical method was applied for adjusting false-discovery rate/multiple testing corrections? A minimum q-value threshold should be clearly stated and included for these metabolites (mouse and human alike).

Response to Reviewers' Comments

We are thankful for your kindly reviewing our manuscript (manuscript entitled "Targeting chondrocytes as a novel strategy for arresting bony fusion in ankylosing spondylitis", Manuscript # NCOMMS-20-40178). According to the reviewers' comments, the manuscript has been substantially revised. Please find below our point-by-point responses.

Reviewer #1 (Remarks to the Author):

The manuscript by Shao and colleagues presents a novel animal model for ankylosing spondylitis and provide also a novel therapeutic approach to treat bone fusion.

-In the overall the manuscript is not well written, and the meaning of several sentences is not clear. E.g.: A novel population of CD4-Cre-expressing hypertrophic chondrocytes was SHP2 deficiency and delayed fusion of growth plate. Premature growth plate fusion by Smo inhibitor sonidegib blocked the AS-like bone disease. The CD4 glycoprotein is expressed as a T cell receptor on the cell membrane. page 8 the paragraph relative to chimera description is quite confusing. The manuscript needs a revision from a native English speaker.

Response: Sorry for the confusion and thank you for your comments. We have carefully revised the manuscript according to the reviewers' comments, and the current manuscript has been revised by a native speaker.

-The presented CKO model as an animal model for AS is not convincing and this fact limits its use. Although the morphological and histological aberrations of the skeleton and joints observed closely recall what observed in AS in human, a more effective comparison with human samples should be provided. Moreover, comparative analysis with human AS samples (or data from literature) at the cellular and molecular level is totally missing, and this may lead to misinterpretations. Whereas the knowledge of the molecular basis of AS is poorly understood yet, autoimmunity and inflammation are two well-known hallmarks of such pathology that should be taken into account in the

development of an animal model to be used in the development of effective targeted therapies. For example, pro-inflammatory profile of CKO model should be provided. Such data should also support the use of Igaratimod (pag. 7).

Response: Thank you for your comments. In the past 6 months, we have analyzed the mechanism of pathological new bone formation in CKO mice at the cellular and molecular level. we confirmed that increased BMP6 expression in chondrocytes promoted the ectopic new bone formation through pSmad1/5 signaling (Fig. 7). At the same time, it is reported that chondrogenesis mediated progression of AS through heterotopic ossification in patients with AS recently. In the early stage of AS, upregulation of pSmad1/5 signaling was detected, suggesting active BMP signaling in ligaments (<https://www.nature.com/articles/s41413-021-00140-6>) (cited in page 4, page 13 and page 15). These results suggest that besides the same skeletal aberrations, the mechanism of pathological new bone formation mediated by chondrogenesis and pSmad1/5 signaling were overlapped in CKO mice and patients with AS.

Fig. 7. Aberrant chondrocytes promoted ectopic new bone formation through

BMP6/Smad1/5 signaling. (A) RT-qPCR analyses of gene expression of chondrocytes after transfected with sh-*Ctrl* and sh-*Ptpn11* respectively. (B) Western blot analyses of chondrocytes of (A). (C) Western blot analysis of phosphorylation of Smad1/5 protein levels in mBMSC stimulated with chondrocytes supernatant for 30 min. (D) *Alizarin Red* staining and quantification of mBMSC cocultured with chondrocytes for 14 days. (E) ALP activity measurement and (F) RT-qPCR analyses of osteogenic marker genes in mBMSC cocultured with chondrocytes for 7 days. (G and H) Immunostaining analysis of pSmad1/5 protein levels in knee joint (G) and intervertebral disc (H). Scale bars: 100 μ m. Data are the mean \pm SEM. * P <0.05, ** P <0.01, *** P <0.001.

In histological analysis, we detected inflammatory infiltration in joint of CKO mice (Fig. 2J and Suppl. Fig. 5). Then, we assessed level of inflammatory factors in serum of CKO mice through ELISA assay. The results showed increased TNF- α , IL-6, IL-17A and CXCL10 in serum of CKO mice. The level of RF and ANA in CKO mice were equal to control mice, which recalled the negative of rheumatoid factor (RF) and antinuclear antibody (ANA) in AS (Suppl. Fig. 6). In the basis of inflammation in CKO mice, we treated CKO mice with Igaratimod. However, Igaratimod just partly rescued the decreased femur BMD and did not alleviate the pathological new bone formation and bony fusion in CKO mice.

Fig. 2

Fig. 2. *H&E* staining images show inflammation, pannus formation and proliferating synoviocytes in knee joints. ($n=5$). Scale bars: 100 μ m.

Supplemental Fig. 5. *Histological analysis shows bone lesions in mature and aged CD4-CKO mice.* (A) H&E staining images show natural structure of wrist from CD4-Ctrl mice and bone deformation and cartilage disorder in wrist of CD4-CKO mice. Scale bars: 200 μ m. (B) H&E staining images of articular cavity of knee joints indicate natural structure of CD4-Ctrl mice and young CD4-CKO mice and cartilage and bone disorder in articular cavity of aged CD4-CKO mice. Scale bars: 100 μ m. (C) Histological images of spines show inflammation and synovial hyperplasia in mature CD4-CKO mice and cartilage and bone disorder in aged CD4-CKO mice. Scale bars: 100 μ m.

Supplemental Fig. 6. *Increased inflammatory cytokines in serum of aged CD4-CKO mice.* Serum of 12-month-old mice CD4-Ctrl and CD4-CKO mice were collected and

the inflammation cytokines were measured by ELISA. (A) Tumor necrosis factor- α (TNF- α). (B) Interleukin-6 (IL-6). (C) Interleukin-17 (IL-17). (D) Interferon γ -induced protein-10 (IP-10, also known as CXCL10). (E) C-X-C motif chemokine ligand 1 (CXCL1). (F) Monocyte chemoattractant protein-1 (MCP-1). (G) Rheumatoid factor (RF). (H) Antinuclear antibody (ANA).

The level of novelty is not very high. One relevant research paper on the role of SHP2 in chondrocytes is missing (<https://doi.org/10.1371/journal.pgen.1004364>). Major findings from the study by Bowen and colleagues are delay in the terminal differentiation of chondrocytes, abnormal organization of chondrocyte maturation zone; significantly increased Hh signaling.

Response: Thank you for your remainder. The role of SHP2 in chondrocytes was same in previous articles and our results, it has been cited in the revised manuscript (cited in page 15-16).

Page 9. please clarify the sentence "...SHP2 deficiency hardly disrupt.....", Is SHP2 conditionally knocked-out only in hypertrophic chondrocyte?

Response: Thank you for your remainder. We traced the origin of SHP2 deletion in chondrocytes by immustaining of Cre. The results indicated Cre expression in proliferating chondrocytes in growth plate and chondrocytes in inner layer of articular cartilage, which could differentiate into pre-hypertrophic and hypertrophic chondrocytes (Suppl. Fig. 10).

Supplemental Fig. 10. *CD4-Cre mediates SHP2 deficiency in proliferating chondrocytes.* Immunohistochemistry analysis of Cre expression knee joint (A) and femoral growth plate (B) showed Cre expressed in proliferating chondrocytes. Scale bars:100 μ m.

Page 11 Which are the basis of the treatment with Hh inhibitor Sonidegib?

Response: Thank you for your comments. Smoothened (Smo) is a transmembrane protein that transduces all Hh signals (page 17). It has been reported that transient inhibition of the Hedgehog pathway by the Smo inhibitor in young mice promoted premature fusion of the epiphyseal plate (growth plate) (page 10). To confirm whether growth plate play a critical role in AS-like bone disease in CD4-CKO mice, we treated young mice with Smo inhibitor sonidegib. The results indicated that transient treatment of sonidegib promoted premature fusion of epiphyseal plate and prevented AS-like bone disease in CKO mice.

It has been reported that SHP2 deficiency increased the expression of Indian hedgehog (*Ihh*) and parathyroid hormone-related protein (*Pthrp*) and caused metachondromatosis, which could be ameliorate by treatment of smoothened (Smo) inhibitor (page 4). In CKO mice, we confirmed that SHP2 deficiency in chondrocytes promoted chondrogenesis and ectopic new bone formation. Therefore, we chose sonidegib to treat bone disease in CKO mice and the results showed sonidegib significantly alleviated chondrogenesis and ectopic new bone formation, as well as osteoporosis in CKO mice. Based on the report that chondrogenesis mediates radiological progression of AS patients (<https://www.nature.com/articles/s41413-021-00140-6>), we propose that blockade of chondrogenesis by sonidegib would be a drug repurposing strategy for AS treatment.

Materials and methods descriptions as well as statistical analysis are adequate

Response: Thank you for your encouragement and giving the above positive comments.

Reviewer #2 (Remarks to the Author):

In this manuscript Dr. Shao and colleagues provide a characterization of CD4-Cre;Ptpn11f/f mice as a model of ankylosing spondylitis. The investigators noted that the mice developed spontaneous kyphosis, abnormal gait, and bony fusion. They examined lineage, hematopoietic cell dependency, inhibitors and metabolic profiles. The inclusion of human data for correlation is an additional strength of the manuscript.

In general the studies are well done. There are several elegant studies including lineage tracing indicating that the pathology is most likely initiated by chondrocytes and the inclusion of lck-cre control experiments is much appreciated as a control for the effects from a T cell population.

The manuscript lacks clarity on a few issues.

Although the authors may not have anticipated the phenotype in retrospect the literature supports their findings: I) Loss of PTPN11 (SHP2) in mice or in patients with PTPN11 mutations (metachondromatosis) has been reported to cause cartilage growths/benign tumors on the bone surface (exostoses) and enchondromas. II) CD4 cre has been described to drive chondrocyte deletion of SOS and ERK (Front. Immunol. 8:343; Front. Immunol. 8:482), which is acknowledged in the results and should be in the introduction or discussion. III) Both SHP2 and ERK have been described as regulating chondrocyte terminal differentiation, growth plate architecture and skeletal cell fates (summarized in PLoS Genet. 2014 May; 10(5): e1004364). This is also buried in the manuscript but the manuscript would be easier to understand if these points were in the introduction. IV) SHP2 deficient chondrocytes can direct endochondromas in trans with other sufficient chondrocytes, so only some of the cartilage needs to be CKD for pathology to occur (PLoS Genet. 2014 May; 10(5): e1004364). V) SHP2 deficiency in fibroblasts lining the bone can cause similar pathology.

Outlining the prior work does not detract from the current extensive analyses and is currently scattered throughout the manuscript making the logic fragmented and difficult

to follow.

Response: Thank you for your comments. In the revised manuscript, we have cited these prior work in the introduction to illustrate the role of SHP2 in chondrocytes (page 4), as well as in the discussion to state the contribution of this article (page 15-16).

The list of AS mouse models the authors should also list the transmembrane TNF models.

Response: Thank you for your comment. We have added the TNF- α transgenic model in the revised manuscript (page 17).

The CD4-CKO mouse at 12 months with splayed feet demonstrates the morbidity with this pathology. Was there premature mortality or paresis associated with the spinal pathology? How frequently did this occur?

Response: The paresis in CD4-CKO mice was attribute to the bony fusion and bone deformation in spine. The paresis could detect in almost all CD4-CKO mice after 11 months. We monitored the mortality rate of mice and the result showed the bone disease led to the death of mice occasionally after 11 months (Suppl. Fig. 1B).

Suppl.Fig.1B

Supplemental Fig. 1. (B) The mortality rate of female CD4-Ctrl and CD4-CKO mice (n=15).

There appears to be bone erosion in the histology which is part of the pathology in human AS and should be specifically mentioned with the TRAP staining.

Response: Thank you for your comment. AS is a type of inflammatory arthritis that affects multiple joints and correction surgery is the final approach to correct spinal deformity. Although we tried our best to collect the tissue of patients with AS, there is not enough bone tissue to assess the osteoclasts in bone of AS patients. We searched the prior work and found previous articles showed increased osteoclasts in the bony surfaces of calcified cartilage and subchondral bone marrow in AS through TRAP staining (cited in page 4).

Figure (e) Tartrate-resistant acid phosphatase (TRAP)-positive cells (red) and (f) quantitative analysis of TRAP-positive osteoclast (red) surface (OCS) per bone surface (BS). The bottom panels show magnified views of the boxed area in the top panels. Scale bar: 100 μ m (top panel); 25 μ m (bottom panel). The data were cited from <https://www.nature.com/articles/s41413-021-00140-6>.

There should be some measures of serum inflammatory cytokines (minimum TNF, KC, IL-6, IP10, MCP-1), and recommend a minimal serologic panel of RF and ANA. The serologic markers are demonstration of classic “seronegative” axial arthritis. This is for characterization of the model in relation to the human condition.

Response: Thank you for your comments. we assessed level of inflammatory factors

in serum of CKO mice through ELISA assay. The results showed increased TNF- α , IL-6, IL-17A and CXCL10 in serum of CKO mice. The level of RF and ANA in CKO mice were equal to control mice, which recalled the negative of rheumatoid factor (RF) and antinuclear antibody (ANA) in AS (Suppl. Fig. 6).

Supplemental Fig. 6. Increased inflammatory cytokines in serum of aged CD4-CKO mice. Serum of 12-month-old mice CD4-Ctrl and CD4-CKO mice were collected and the inflammation cytokines were measured by ELISA. (A) Tumor necrosis factor- α (TNF- α). (B) Interleukin-6 (IL-6). (C) Interleukin-17 (IL-17). (D) Interferon γ -induced protein-10 (IP-10, also known as CXCL10). (E) C-X-C motif chemokine ligand 1 (CXCL1). (F) Monocyte chemoattractant protein-1 (MCP-1). (G) Rheumatoid factor (RF). (H) Antinuclear antibody (ANA).

The inhibition of pathology with a smoothed antagonist speaks further to a cartilage associated pathology as IHH has been implicated in cartilage tumors and aberrant growth. However, the partial effect of the smoothed inhibitor does not warrant the conclusion that “chondroma plays a trivial role in AS-like bone lesions in CD4-CKO mice.” as this is likely a complex process, the treatment at 2 months was effective but less so at 6 months which allowed time for additional processes, and the CD4-CKO chondrocytes are likely acting in trans on the other chondrocytes. In addition, the scores in Supplemental figure 12 for the control group are relatively low compared to the other experiments and the difference would have appeared to be greater. The conclusion for

treatment of the younger group: “Taken together, the data indicated that induction of premature fusion of growth plates prevent the AS-like bone disease in CD4-CKO mice” is not precise as the data reflect that the premature fusion of the growth plate is a marker of the mechanisms associated with the formation of chondromas, and abnormal bone formation and fusion. Fusion of the growth plate in itself is unlikely to arrest all of the pathologies.

Response: Thank you for your comments. As you suggested, SHP2-deficient chondrocytes facilitated the accumulation of wild-type chondrocytes in chondromas of CD4-CKO mice (Fig. 4F).

Fig. 4F

Fig. 4. (F) Immunofluorescence analysis of chondroma of the tibia in 7-month-old CD4-CKO; Rosa26-mTmG mice show GFP⁺ differentiated chondrocytes in growth plate and articular cartilage of mature CD4-CKO; Rosa26-mTmG mice ($n=5$). Scale bars, 100 μ m.

There were more GFP⁺ differentiated chondrocytes in young CD4-CKO mice than mature CD4-CKO mice. However, the chondromas and bone deformation were detected in mature CD4-CKO mice, rather than young CD4-CKO mice. It was different from the metachondromatosis and exostoses in *Col2a1-Cre;Ptpn11^{fl/fl}* mice and *Ctsk-Cre;Ptpn11^{fl/fl}* mice, which occurred within 2 months. These results suggested that CD4-Cre-mediated conditional SHP2 deficiency in chondrocytes is insufficient to lead to AS-like bone disease in mice, and that other factors, such as maturation or aging, may be involved in the process. The bone deformation in CD4-CKO mice focused on axial and peripheral joints, where the epiphyseal plate existed. In addition, the chondromas was abolished by premature of growth plate completely. It suggested that aberrant chondrocytes in growth plate promoted the formation of chondromas in mature CD4-CKO mice. Although the low-dose sonidegib did not alleviated the bone disease, we treated the mature CD4-CKO mice with high-dose sonidegib. High-dose sonidegib significantly alleviated the bone deformation in CD4-CKO mice, which would benefit from the double inhibition of chondrogenesis in growth plate and chondromas of sonidegib (Fig. 8).

Fig. 8. Targeting chondrocytes retarded AS progression in CD4-CKO mice. 7-month-old CD4-CKO mice and CD4-Cre littermates were orally gavaged with Smo inhibitor, sonidegib (100 mg/kg) for 4 months. (A) Pathological score ($n=6$). (B) Gross images. (C) X-ray images. (B and C) Scale bars: 1 cm. (D) Femur bone mineral density (BMD). (E) H&E staining of knee joints. (F) SOFG staining of knee joints. (G and H) μ -CT radiographs of knee joint (G) and spine (H) of 11-month-old mice. Scale bars: 2.5 cm. (I) Immunohistochemical staining of pSmad1/5 protein levels in knee joint of 11-month-old mice. (E, F and I) Scale bars: 100 μ m. Data are the mean \pm SEM. ** $P<0.01$, *** $P<0.001$. (J) The mechanism of AS-like bone disease in CD4-CKO mice.

The choice of Igaratimod as a treatment agent should be discussed as this is not in the treatment regimens for AS (as outlined by the authors). This treatment abrogated weight loss and may have had an effect on TNF in this model. The other treatments had BMD checked. Was it also checked for this treatment group?

Response: Thank you for your comments. The choice of Igaratimod has been discussed in the revised manuscript (page 16). The measurement of femur BMD has been shown in the revised manuscript (Suppl. Fig. 8). the result showed Igaratimod reduced the bone loss in CD4-CKO mice, suggesting inflammation promoted the bone loss in CD4-CKO mice.

Supplemental Fig. 8. Anti-rheumatic drug Igaratimod does not improve AS-like bone disease in CD4-CKO mice. 6-month-old CD4-CKO mice were treatment with 100 mg/kg Igaratimod and the body weights and pathological scores were monitored for 16 weeks. (A) Body weights of CD4-Ctrl and CD4-CKO mice treated with Igaratimod or not. (B) Pathological scores of bone disease in CD4-Ctrl, CD4-CKO mice and CD4-CKO mice treated with Igaratimod ($n=5$). (C-D) Representative X-ray images (C) and femoral BMD (D) of CD4-CKO mice treated with Igaratimod for 16 weeks or not and CD4-Ctrl littermates. Scale bar: 1 cm. Data are the mean \pm SEM. * $P<0.05$, ** $P<0.01$, *** $P<0.001$.

Please review the manuscript for typographical errors and a few are listed:

Osteoalleosis, osteophyte accumulation, and osteoporosis in CD4-CKO mice but not in Sonidegib-treated CD4-CKO mice (Figure 5G and H)-this sentence is missing a verb.

The metabolites were significant differences- do the authors mean there were significant differences between metabolites or the metabolites themselves were different?

Figure 8 panel B uirc should be uric

Supplemental figure 15

Suramroin change to Suramrin

Response: We appreciate for your valuable comment and sorry for our carelessness. The label mistake has been corrected in the revised manuscript. Thank you so much for your comments.

Reviewer #3 (Remarks to the Author):

In the present work, the authors reported that deletion of SHP2 in CD4⁺ cells caused age-related AS-like bone disease which had similar pathology to AS patients. This phenotype is dependent of Shp2 deficiency in hypertrophic chondrocytes but not in CD4⁺ T cells. Mechanistically, SHP2-deficiency led to the increased purine metabolism in chondrocytes, thus promoting bone deformation through paracrine mechanisms. In general, the findings in this study are novel, and may provide a potential strategy for arresting new bone formation and radiological progression of AS. However, there are still some concerns need to be addressed to strengthen the present work.

1. The authors came to the conclusion that SHP2-deficient T cells were unlikely the effector cells that induce the AS-like bone lesions in CD4-CKO mice by adoptive transfer experiment. However, it is still possible that Shp2 deletion in CD4⁺ T cells (or other CD4⁺ immune cells, such as a subset of NK cells) caused immune abnormality in CD4-CKO mice, which indirectly affected the function of Shp2-deficient chondrocytes but not Shp2-sufficient chondrocytes. In other words, both CD4⁺ T cell-

and chondrocyte-expressed *Shp2* are responsible for the phenotype observed in this work. It is best to design more experiments to exclude this possibility.

Response: Thank you for your comments. we performed many experiments after T cell adoptive transfer experiment to clarify the role of SHP2-deficient T cells in the bone disease of CD4-CKO mice. There are no any bone deformation in Lck-Cre;SHP2^{fl/fl} mice, in which SHP2 was conditional deleted in T lymphocytes. It confirmed that SHP2-deficient T cells cannot induce bone deformation directly. Moreover, we performed bone marrow transplant and the bone deformation was equivalent in the irradiated CD4-CKO mice received CD4-Ctrl bone marrow cells and the irradiated CD4-CKO mice received CD4-CKO bone marrow cells. These results suggested SHP2-deficient T cells were independent of bone disease in CD4-CKO mice.

Fig. 3. (D-F) Gross images (D), X-ray images (E) and Femoral BMD (F) of Lck-CKO mice and Lck-Ctrl littermates. ($n=5$). (G-K) CD4-Ctrl (WT) and CD4-CKO (KO) mice at 4 months were lethally irradiated followed by transferring with bone-marrow cells. Irradiated CD4-CKO mice transferred with WT bone-marrow cells were labeled as WT-KO, and so on. Pathological scores (G), Gross images (H) and Radiographs (I) of 12-month-old chimeras. (C-E, H-I) Scale bars: 1 cm. (J-K) H&E and SOFG staining images of chimeras. Scale bars: 200 μ m. Data are the mean \pm SEM. * $P<0.05$. ** $P<0.01$. *** $P<0.001$.

2. In the pathogenesis of AS, which cells are the main source of purine or purine

metabolites?

3.The authors observed that SHP2-knockdown in chondrocytes resulted in the increased production of cyclic AMP, hypoxanthine, and xanthine. What is the possible underlying mechanism(s)?

4.In the final section of the Results, the authors only found an increased purine metabolites in the serum of patients with AS. It is insufficient to come to the conclusion that purine metabolism facilitates the radiological progression of AS through paracrine signaling. The enhanced level of purine metabolites might be secondary to the pathogenesis of AS, not the cause. The authors should provide more evidence (or at least cite some previous papers) to support their conclusion.

Response: Thank you for your comments. Based on the discovery of mechanism of pathological new bone formation and new treatment for arresting bony fusion in AS, we removed the results of paracrine pathway mediated by purine in CD4-CKO mice in the revised manuscript. We still very appreciate all your comments about purine, which will assist our subsequent study.

5.It has been reported that Shp2 deficient in CD4+ Cells causes cartilage malfunction without affecting T Cell development and functions. (“Ptpn11 Deletion in CD4+ Cells Does Not Affect T Cell Development and Functions but Causes Cartilage Tumors in a T Cell-Independent Manner. Front. Immunol.2017”). The authors should put more discussion on this and describe how this study relates to your present work.

Response: Thank you for your comment. we have cited these prior work and discussed in the revised manuscript (cited in page 9 and page16).

Reviewer #4 (Remarks to the Author):

The comments provided for this peer-review is heavily focused on the reporting of non-targeted metabolomics performed alongside a number of immunological and gross

phenotypic parameters. Notably, non-targeted metabolomics was utilized to support the findings for purine metabolites as the responsible metabolic pathway to facilitate AS-like bone disease in CD4-CKO mice AND that was differentiating AS patients from healthy controls. A number of major additional points of clarification are needed to support the approach and the results interpretations prior to acceptance for publication.

1. The rationale and justification for limiting the non-targeted metabolomics approach to the targeted analysis of purine nucleotides is needed. There are likely to be a number of other nucleotide as well as other metabolic pathways classes involved in distinguishing this phenotype that are not mentioned. Figure 8 was incomplete as the multiple other metabolic pathways identified that distinguish human patients from healthy controls were not included for enhanced understanding of the relative importance of selected metabolites.

2. What are the total number of metabolites that were significantly different for both animal and human datasets (across all metabolic pathways examined using metabolomics)? The data as shown does not support that the differential metabolic productions (DMPs) were primarily important to purine metabolism.

3. The methods section states use of both plasma and serum, and this should be clearly indicated which was applied for the data presented in mice and human. Additionally, if it is plasma, then the anticoagulant used during blood collection should also be mentioned for reproducibility.

4. There are no method details provided regarding the process for metabolite identification (assuming library database is available for curation?). Were commercial standards used for library, what is the level of confidence for confirmation of the 43 metabolite identifications that varied significantly in the serum between the mature CD4-CKO mice and CD4-Ctrl littermates? Was the same method for identification used for the human sera?

5. How were the raw non-targeted metabolomics data analyzed, was this prior to metabolite identification, were the fold differences on relative abundances normalized or median scaled? What does the scale represent for Figure 7D? Assuming all statistics were applied on fold differences between groups or were relative abundances used in

some cases?

- 6. What are the group comparisons for pathway enrichments shown in Figure 7C and 7F, and what are the total number of metabolites included in these pathway analyses. The analysis and results should include the relative importance of the purine metabolic pathway enrichment to other scores for nucleotide or metabolic pathways. Inclusion of the additional metabolic pathways affected will bolster the findings for a role in purine metabolism as currently this appears to be a targeted analysis using a non-targeted metabolomics dataset.*
- 7. The description of the unsupervised cluster and pathway analysis was incomplete. Additionally, no PCA or OPLS-DA was provided for animals as was included for humans. Was there a separation or heat-map completed for more pathways?*
- 8. What statistical method was applied for adjusting false-discovery rate/multiple testing corrections? A minimum q-value threshold should be clearly stated and included for these metabolites (mouse and human alike).*

Response: Thank you for your comments. In order to add the mechanism of pathological new bone formation, we removed the results of paracrine pathway mediated by purine in CD4-CKO mice in the revised manuscript. We greatly appreciate all your comments about metabolomics, which will assist our subsequent research.

Thank you so much for the editor and reviewers' precious comments and suggestions.

Reviewers' Comments:

Reviewer #1:

Remarks to the Author:

Most of my concerns have been addressed by authors and the revised manuscript has been notably improved.

Reviewer #2:

Remarks to the Author:

The revisions substantially improve the manuscript and the writing is easier to follow.

Reviewer #3:

Remarks to the Author:

In this revised ms, authors have satisfactorily addressed most of my concerns.

Response to Reviewers' Comments

We are thankful for your kindly reviewing our manuscript (manuscript entitled "Targeting chondrocytes as a novel strategy for arresting bony fusion in ankylosing spondylitis", Manuscript # NCOMMS-20-40178). According to the reviewers' comments, the manuscript has been substantially revised. Please find below our point-by-point responses.

Reviewer #1 (Remarks to the Author):

The manuscript by Shao and colleagues presents a novel animal model for ankylosing spondylitis and provide also a novel therapeutic approach to treat bone fusion.

-In the overall the manuscript is not well written, and the meaning of several sentences is not clear. E.g.: A novel population of CD4-Cre-expressing hypertrophic chondrocytes was SHP2 deficiency and delayed fusion of growth plate. Premature growth plate fusion by Smo inhibitor sonidegib blocked the AS-like bone disease. The CD4 glycoprotein is expressed as a T cell receptor on the cell membrane. page 8 the paragraph relative to chimera description is quite confusing. The manuscript needs a revision from a native English speaker.

Response: Sorry for the confusion and thank you for your comments. We have carefully revised the manuscript according to the reviewers' comments, and also have re-scrutinized to improve the writing by a native speaker.

-The presented CKO model as an animal model for AS is not convincing and this fact limits its use. Although the morphological and histological aberrations of the skeleton and joints observed closely recall what observed in AS in human, a more effective comparison with human samples should be provided. Moreover, comparative analysis with human AS samples (or data from literature) at the cellular and molecular level is totally missing, and this may lead to misinterpretations. Whereas the knowledge of the molecular basis of AS is poorly understood yet, autoimmunity and inflammation are two well-known hallmarks of such pathology that should be taken into account in the development of an animal model to be used in the development of effective targeted therapies. For example, pro-inflammatory profile of CKO model should be provided.

Such data should also support the use of Igaratimod (pag. 7).

Response: Thank you for your comments. In the past 6 months, we have analyzed the mechanism of pathological new bone formation in CKO mice at the cellular and molecular level. We confirmed that increased BMP6 expression in chondrocytes promoted the ectopic new bone formation through pSmad1/5 signaling (Fig. 7). At the same time, it is reported that chondrogenesis mediated progression of AS through heterotopic ossification in patients with AS recently. In the early stage of AS, upregulation of pSmad1/5 signaling was detected, suggesting active BMP signaling in ligaments (<https://www.nature.com/articles/s41413-021-00140-6>) (cited in page 4, page 13 and page 15). These results suggest that besides the same skeletal aberrations, the mechanism of pathological new bone formation mediated by chondrogenesis and pSmad1/5 signaling were overlapped in CKO mice and patients with AS.

Fig. 7. Aberrant chondrocytes promote ectopic new bone formation through BMP/Smad1/5 signaling. (a) RT-qPCR analyses of gene expression of chondrocytes after transfected with sh-Ctrl and sh-Ptpn11 respectively. ($n=3$). (b) Western blot

analyses of chondrocytes of (a). (c) western blot analysis of phosphorylation of Smad1/5 protein levels in mBMSC stimulated with chondrocytes supernatant for 30 min. (d) *Alizarin Red staining and quantification of mBMSC cocultured with chondrocytes for 14 days.* (n=5). (e) *ALP activity measurement* (n=6) and (f) *RT-qPCR analyses* (n=3) of osteogenic marker genes in mBMSC cocultured with chondrocytes for 7 days. (g and h) *Immunostaining analysis of pSmad1/5 protein levels in knee joint* (g) and *intervertebral disc* (h). Scale bars: 100 μ m. (a, d, e, f) Data were presented as mean \pm SEM. * $P < 0.05$, ** $P < 0.01$, *** $P < 0.001$, determined by two-tailed Student's *t*-test. (b-d, g, h) Data are representative of three independent biological replicates.

In histological analysis, we detected inflammatory infiltration in joint of CKO mice (Fig. 2J and Suppl. Fig. 5). Then, we assessed level of inflammatory factors in serum of CKO mice through ELISA assay. The results showed increased TNF- α , IL-6, IL-17A and CXCL10 in serum of CKO mice. The level of RF and ANA in CKO mice were equal to control mice, which recalled the negative of rheumatoid factor (RF) and antinuclear antibody (ANA) in AS (Suppl. Fig. 6). In the basis of inflammation in CKO mice, we treated CKO mice with Igaratimod. However, Igaratimod just partly rescued the decreased femur BMD and did not alleviate the pathological new bone formation and bony fusion in CKO mice.

Fig. 2

Fig. 2. (j) H&E staining images show inflammation, pannus formation and proliferating synoviocytes in knee joints. Scale bars: 100 μ m. Data are representative of five independent biological replicates.

Supplementary Fig. 5. Histological analysis shows bone lesions in mature and aged CD4-CKO mice. (a) H&E staining images show natural structure of wrist from CD4-Ctrl mice and bone deformation and cartilage disorder in wrist of CD4-CKO mice. Scale bars: 200 μm . (b) H&E staining images of articular cavity of knee joints indicate natural structure of CD4-Ctrl mice and young CD4-CKO mice and cartilage and bone disorder in articular cavity of aged CD4-CKO mice. Scale bars: 100 μm . (c) Histological images of spines show inflammation and synovial hyperplasia in mature CD4-CKO mice and cartilage and bone disorder in aged CD4-CKO mice. Scale bars: 100 μm . (a-c) Data are representative of three independent biological replicates.

Supplementary Fig. 6. Increased inflammatory cytokines in serum of aged CD4-CKO mice. Serum of 12-month-old mice CD4-Ctrl and CD4-CKO mice were collected and the inflammation cytokines were measured by ELISA. (a) Tumor necrosis factor- α (TNF- α). (b) Interleukin-6 (IL-6). (c) Interleukin-17A (IL-17A). (d) Interferon γ -induced protein-10 (IP-10, also known as CXCL10). (e) C-X-C motif chemokine ligand 1 (CXCL1). (f) Monocyte chemoattractant protein-1 (MCP-1). (g) Rheumatoid factor (RF). (h) Antinuclear antibody (ANA). (a-h) ($n=6$). Data were presented as mean \pm SEM. * $P<0.05$, ** $P<0.01$, *** $P<0.001$, determined by two-tailed Student's t -test.

The level of novelty is not very high. One relevant research paper on the role of SHP2 in chondrocytes is missing (<https://doi.org/10.1371/journal.pgen.1004364>). Major findings from the study by Bowen and colleagues are delay in the terminal differentiation of chondrocytes, abnormal organization of chondrocyte maturation zone; significantly increased Hh signaling.

Response: Thank you for your remainder. The role of SHP2 in chondrocytes was same in previous articles and our results, it has been cited in the revised manuscript (cited in page 15-16).

Page 9. please clarify the sentence "...SHP2 deficiency hardly disrupt.....", Is SHP2 conditionally knocked-out only in hypertrophic chondrocyte?

Response: Thank you for your remainder. We traced the origin of SHP2 deletion in

chondrocytes by immunostaining of Cre. The results indicated Cre expression in proliferating chondrocytes in growth plate and chondrocytes in inner layer of articular cartilage, which could differentiate into pre-hypertrophic and hypertrophic chondrocytes (Suppl. Fig. 10).

Supplementary Fig. 10. CD4-Cre mediates SHP2 deficiency in proliferating chondrocytes. (a-b) Immunohistochemistry analysis of Cre expression knee joint (a) and femoral growth plate (b) show Cre expressed in proliferating chondrocytes. Scale bars:100 μm . Data are representative of three independent experiments.

Page 11 Which are the basis of the treatment with Hh inhibitor Sonidegib?

Response: Thank you for your comments. Smoothened (Smo) is a transmembrane protein that transduces all Hh signals (page 17). It has been reported that transient inhibition of the Hedgehog pathway by the Smo inhibitor in young mice promoted premature fusion of the epiphyseal plate (growth plate) (page 10). To confirm whether growth plate play a critical role in AS-like bone disease in CD4-CKO mice, we treated young mice with Smo inhibitor sonidegib. The results indicated that transient treatment of sonidegib promoted premature fusion of epiphyseal plate and prevented AS-like bone disease in CKO mice.

It has been reported that SHP2 deficiency increased the expression of Indian hedgehog (*Ihh*) and parathyroid hormone-related protein (*Pthrp*) and caused metachondromatosis, which could be ameliorate by treatment of smoothened (Smo) inhibitor (page 4). In CKO mice, we confirmed that SHP2 deficiency in chondrocytes

promoted chondrogenesis and ectopic new bone formation. Therefore, we chose sonidegib to treat bone disease in CKO mice and the results showed sonidegib significantly alleviated chondrogenesis and ectopic new bone formation, as well as osteoporosis in CKO mice. Based on the report that chondrogenesis mediates radiological progression of AS patients (<https://www.nature.com/articles/s41413-021-00140-6>), we would like to propose that sonidegib could be the drug candidate for AS treatment.

Materials and methods descriptions as well as statistical analysis are adequate

Response: Thank you for reading our paper carefully and giving the above positive comments.

Reviewer #2 (Remarks to the Author):

In this manuscript Dr. Shao and colleagues provide a characterization of CD4-Cre;Ptpn11f/f mice as a model of ankylosing spondylitis. The investigators noted that the mice developed spontaneous kyphosis, abnormal gait, and bony fusion. They examined lineage, hematopoietic cell dependency, inhibitors and metabolic profiles. The inclusion of human data for correlation is an additional strength of the manuscript.

In general the studies are well done. There are several elegant studies including lineage tracing indicating that the pathology is most likely initiated by chondrocytes and the inclusion of lck-cre control experiments is much appreciated as a control for the effects from a T cell population.

The manuscript lacks clarity on a few issues.

Although the authors may not have anticipated the phenotype in retrospect the literature supports their findings: 1) Loss of PTPN11 (SHP2) in mice or in patients with PTPN11 mutations (metachondromatosis) has been reported to cause cartilage growths/benign

tumors on the bone surface (exostoses) and enchondromas. II) CD4 cre has been described to drive chondrocyte deletion of SOS and ERK (Front. Immunol. 8:343; Front. Immunol. 8:482), which is acknowledged in the results and should be in the introduction or discussion. III) Both SHP2 and ERK have been described as regulating chondrocyte terminal differentiation, growth plate architecture and skeletal cell fates (summarized in PLoS Genet. 2014 May; 10(5): e1004364). This is also buried in the manuscript but the manuscript would be easier to understand if these points were in the introduction. IV) SHP2 deficient chondrocytes can direct endochondromas in trans with other sufficient chondrocytes, so only some of the cartilage needs to be CKD for pathology to occur (PLoS Genet. 2014 May; 10(5): e1004364). V) SHP2 deficiency in fibroblasts lining the bone can cause similar pathology.

Outlining the prior work does not detract from the current extensive analyses and is currently scattered throughout the manuscript making the logic fragmented and difficult to follow.

Response: Thank you for your comments. In the revised manuscript, we have cited these prior work in the introduction to illustrate the role of SHP2 in chondrocytes (page 4), as well as in the discussion to state the contribution of this article (page 15-16).

The list of AS mouse models the authors should also list the transmembrane TNF models.

Response: Thank you for your comment. We have added the TNF- α transgenic model in the revised manuscript (page 17).

The CD4-CKO mouse at 12 months with splayed feet demonstrates the morbidity with this pathology. Was there premature mortality or paresis associated with the spinal pathology? How frequently did this occur?

Response: The paresis in CD4-CKO mice was attribute to the bony fusion and bone deformation in spine. The paresis could detect in almost all CD4-CKO mice after 11 months. We monitored the mortality rate of mice and the result showed the bone disease led to the death of mice occasionally after 11 months (Suppl. Fig. 1B).

Supplementary Fig. 1

Supplementary Fig. 1. (b) The mortality rate of female CD4-Ctrl and CD4-CKO mice (n=10)

There appears to be bone erosion in the histology which is part of the pathology in human AS and should be specifically mentioned with the TRAP staining.

Response: Thank you for your comment. AS is a type of inflammatory arthritis that affects multiple joints and correction surgery is the final approach to correct spinal deformity. Although we tried our best to collect the tissue of patients with AS, there is not enough bone tissue to assess the osteoclasts in bone of AS patients. We searched the prior work and found previous articles showed increased osteoclasts in the bony surfaces of calcified cartilage and subchondral bone marrow in AS through TRAP staining (cited in page 4).

Figure (e) Tartrate-resistant acid phosphatase (TRAP)-positive cells (red) and (f) quantitative analysis of TRAP-positive osteoclast (red) surface (OCS) per bone surface

(BS). The bottom panels show magnified views of the boxed area in the top panels. Scale bar: 100 μm (top panel); 25 μm (bottom panel). (data cited from <https://www.nature.com/articles/s41413-021-00140-6>)

There should be some measures of serum inflammatory cytokines (minimum TNF, KC, IL-6, IP10, MCP-1), and recommend a minimal serologic panel of RF and ANA. The serologic markers are demonstration of classic “seronegative” axial arthritis. This is for characterization of the model in relation to the human condition.

Response: Thank you for your comments. we assessed level of inflammatory factors in serum of CKO mice through ELISA assay. The results showed increased TNF- α , IL-6, IL-17A and CXCL10 in serum of CKO mice. The level of RF and ANA in CKO mice were equal to control mice, which recalled the negative of rheumatoid factor (RF) and antinuclear antibody (ANA) in AS (Suppl. Fig. 6).

Supplementary Fig. 6. Increased inflammatory cytokines in serum of aged CD4-CKO mice. Serum of 12-month-old mice CD4-Ctrl and CD4-CKO mice were collected and the inflammation cytokines were measured by ELISA. (a) Tumor necrosis factor- α (TNF- α). (b) Interleukin-6 (IL-6). (c) Interleukin-17A (IL-17A). (d) Interferon γ -induced protein-10 (IP-10, also known as CXCL10). (e) C-X-C motif chemokine ligand 1 (CXCL1). (f) Monocyte chemoattractant protein-1 (MCP-1). (g) Rheumatoid factor (RF). (h) Antinuclear antibody (ANA). (a-h) ($n=6$). Data were presented as mean \pm

SEM. * $P < 0.05$, ** $P < 0.01$, *** $P < 0.001$, determined by two-tailed Student's t -test.

The inhibition of pathology with a smoothed antagonist speaks further to a cartilage associated pathology as IHH has been implicated in cartilage tumors and aberrant growth. However the partial effect of the smoothed inhibitor does not warrant the conclusion that “chondroma plays a trivial role in AS-like bone lesions in CD4-CKO mice.” as this is likely a complex process, the treatment at 2 months was effective but less so at 6 months which allowed time for additional processes, and the CD4-CKO chondrocytes are likely acting in trans on the other chondrocytes. In addition the scores in Supplemental figure 12 for the control group are relatively low compared to the other experiments and the difference would have appeared to be greater. The conclusion for treatment of the younger group: “Taken together, the data indicated that induction of premature fusion of growth plates prevent the AS-like bone disease in CD4-CKO mice” is not precise as the data reflect that the premature fusion of the growth plate is a marker of the mechanisms associated with the formation of chondromas, and abnormal bone formation and fusion. Fusion of the growth plate in itself is unlikely to arrest all of the pathologies.

Response: Thank you for your comments. As you said, SHP2-deficient chondrocytes facilitated the accumulation of wild-type chondrocytes in chondromas of CD4-CKO mice (Fig. 4F).

Fig. 4

f

Fig. 4. (f) Immunofluorescence analysis of chondroma of the tibia in 7-month-old CD4-CKO; Rosa26-mTmG mice show GFP⁺ differentiated chondrocytes in growth plate and articular cartilage of mature CD4-CKO; Rosa26-mTmG mice. Scale bars, 100 μ m. Data are representative of three independent experiments.

There were more GFP⁺ differentiated chondrocytes in young CD4-CKO mice than mature CD4-CKO mice. However, the chondromas and bone deformation were detected in mature CD4-CKO mice, rather than young CD4-CKO mice. It was different from the metachondromatosis and exostoses in *Col2a1-Cre; Ptpn11^{fl/fl}* mice and *Ctsk-Cre; Ptpn11^{fl/fl}* mice, which occurred within 2 months. These results suggested that CD4-Cre-mediated conditional SHP2 deficiency in chondrocytes is insufficient to lead to AS-like bone disease in mice, and that other factors, such as maturation or aging, may be involved in the process. The bone deformation in CD4-CKO mice focused on axial and peripheral joints, where the epiphyseal plate existed. In addition, the chondromas was abolished by premature of growth plate completely. It suggested that aberrant chondrocytes in growth plate promoted the formation of chondromas in mature CD4-CKO mice. Although the low-dose sonidegib did not alleviated the bone disease, we treated the mature CD4-CKO mice with high-dose sonidegib. High-dose sonidegib significantly alleviated the bone deformation in CD4-CKO mice, which would benefit from the double inhibition of chondrogenesis in growth plate and chondromas of sonidegib (Fig. 8).

Fig. 8. Targeting chondrocyte retards AS progression in CD4-CKO mice. 7-month-old CD4-CKO mice and CD4-Cre littermates were orally gavaged with Smo inhibitor, sonidegib (100 mg/kg) for 4 months. (a) Pathological score ($n=6$). (b) Gross images. (c) X-ray images. (b and c) Scale bars: 1 cm. (d) Femur bone mineral density (BMD). ($n=6$) (e) H&E staining of knee joints. (f) SOFG staining of knee joints. (g and h) μ -CT radiographs of knee joint (g) and spine (h) of 11-month-old mice. Scale bars: 2.5 cm. (i) Immunohistochemical staining of pSmad1/5 protein levels in knee joint of 11-month-

old mice. (e, f, i) Scale bars: 100 μ m. (a, d) Data were presented as mean \pm SEM. ** $P < 0.01$, *** $P < 0.001$, determined by two-tailed Student's t -test. (b, c, e-i) Data are representative of three independent biological replicates. (j) The mechanism of AS-like bone disease in CD4-CKO mice.

The choice of Igaratimod as a treatment agent should be discussed as this is not in the treatment regimens for AS (as outlined by the authors). This treatment abrogated weight loss and may have had an effect on TNF in this model. The other treatments had BMD checked. Was it also checked for this treatment group?

Response: Thank you for your comments. The choice of Igaratimod has been discussed in the revised manuscript (page 16). The measurement of femur BMD has been shown in the revised manuscript (Suppl. Fig. 8). The result showed Igaratimod reduced the bone loss in CD4-CKO mice, suggesting inflammation promoted the bone loss in CD4-CKO mice.

Supplementary Fig. 8. Anti-rheumatic drug Igaratimod does not improve AS-like bone disease in CD4-CKO mice. 6-month-old CD4-CKO mice were treatment with 100 mg/kg Igaratimod and the body weights and pathological scores were monitored for 16 weeks. (a) Body weights of CD4-Ctrl and CD4-CKO mice treated with Igaratimod or not. (b) Pathological scores of bone disease in CD4-Ctrl, CD4-CKO mice and CD4-CKO mice treated with Igaratimod. (c-d) Representative X-ray images (c)

and femoral BMD. (d) of CD4-CKO mice treated with Igaratimod for 16 weeks or not and CD4-Ctrl littermates. (c) Scale bar: 1 cm. (a, b, d) ($n=6$). Data were presented as mean \pm SEM. * $P<0.05$, ** $P<0.01$, *** $P<0.001$, determined by two-tailed Student's t -test. (c) Data are representative of three independent biological replicates.

Please review the manuscript for typographical errors and a few are listed:

Osteoalleosis, osteophyte accumulation, and osteoporosis in CD4-CKO mice but not in Sonidegib-treated CD4-CKO mice (Figure 5G and H)-this sentence is missing a verb.

The metabolites were significant differences- do the authors mean there were significant differences between metabolites or the metabolites themselves were different?

Figure 8 panel B uirc should be uric

Supplemental figure 15

Suramroin change to Suramrin

Response: We appreciate for your valuable comment and sorry for our carelessness. The label mistake has been corrected in the revised manuscript. Thank you so much for your careful check.

Reviewer #3 (Remarks to the Author):

In the present work, the authors reported that deletion of SHP2 in CD4⁺ cells caused age-related AS-like bone disease which had similar pathology to AS patients. This phenotype is dependent of Shp2 deficiency in hypertrophic chondrocytes but not in CD4⁺ T cells. Mechanistically, SHP2-deficiency led to the increased purine metabolism in chondrocytes, thus promoting bone deformation through paracrine mechanisms. In general, the findings in this study are novel, and may provide a potential strategy for arresting new bone formation and radiological progression of AS. However, there are still some concerns need to be addressed to strengthen the present work.

1.The authors came to the conclusion that SHP2-deficient T cells were unlikely the effector cells that induce the AS-like bone lesions in CD4-CKO mice by adoptive

transfer experiment. However, it is still possible that *Shp2* deletion in CD4+ T cells (or other CD4+ immune cells, such as a subset of NK cells) caused immune abnormality in CD4-CKO mice, which indirectly affected the function of *Shp2*-deficient chondrocytes but not *Shp2*-sufficient chondrocytes. In other words, both CD4+ T cell- and chondrocyte-expressed *Shp2* are responsible for the phenotype observed in this work. It is best to design more experiments to exclude this possibility.

Response: Thank you for your comments. we performed many experiments after T cell adoptive transfer experiment to clarify the role of SHP2-deficient T cells in the bone disease of CD4-CKO mice. There are no any bone deformation in Lck-Cre;SHP2^{fl/fl} mice, in which SHP2 was conditional deleted in T lymphocytes. It confirmed that SHP2-deficient T cells cannot induce bone deformation directly. Moreover, we performed bone marrow transplant and the bone deformation was equivalent in the irradiated CD4-CKO mice received CD4-Ctrl bone marrow cells and the irradiated CD4-CKO mice received CD4-CKO bone marrow cells. These results showed SHP2-deficient T cells were independent of bone disease in CD4-CKO mice.

Fig. 3

Fig. 3. (d-f) Gross images (d), X-ray images (e) and Femoral BMD (f) of Lck-CKO mice and Lck-Ctrl littermates. ($n=5$). (g-k) CD4-Ctrl (WT) and CD4-CKO (KO) mice at 4 months were lethally irradiated followed by transferring with bone-marrow cells.

Irradiated CD4-CKO mice transferred with WT bone-marrow cells were labeled as WT-KO, and so on. Pathological scores ($n=10$) (g), Gross images (h) and Radiographs (i) of 12-month-old chimeras. (c-e, h-i) Scale bars: 1 cm. (j-k) H&E and SOFG staining images of chimeras. Scale bars: 200 μm . (a, b, f, g) Data were presented as mean \pm SEM. $*P<0.05$, $**P<0.01$, $***P<0.001$, determined by two-tailed Student's t -test. (c-e, h-k) Data are representative of three independent biological replicates.

2. In the pathogenesis of AS, which cells are the main source of purine or purine metabolites?

Response: Thank you for your comments. Targeting paracrine pathway in chondrocyte just partially inhibited the process of bone deformation and the mechanism of purine upregulation is unclear. In the past 6 months, we clarified chondrogenesis mediated pathological new bone formation through BMP6/pSmad1/5 signaling and confirmed that drug sonidegib retarded ectopic new bone formation and bony fusion in mice through directly targeting chondrogenesis. To add the mechanism of pathological new bone formation (Fig. 7 and Fig. 8), we removed the results of paracrine pathway mediated by purine in CD4-CKO mice in the revised manuscript. We still very appreciate all your comments about metabolomics, which will assist our subsequent research.

3. The authors observed that SHP2-knockdown in chondrocytes resulted in the increased production of cyclic AMP, hypoxanthine, and xanthine. What is the possible underlying mechanism(s)?

Response: Thank you for your comments. Targeting paracrine pathway in chondrocyte just partially inhibited the process of bone deformation and the mechanism of purine upregulation is unclear. In the past 6 months, we clarified chondrogenesis mediated

pathological new bone formation through BMP6/pSmad1/5 signaling and confirmed that drug sonidegib retarded ectopic new bone formation and bony fusion in mice through directly targeting chonrogenesis. To add the mechanism of pathological new bone formation (Fig. 7 and Fig. 8), we removed the results of paracrine pathway mediated by purine in CD4-CKO mice in the revised manuscript. We still very appreciate all your comments about metabolomics, which will assist our subsequent research.

4. In the final section of the Results, the authors only found an increased purine metabolites in the serum of patients with AS. It is insufficient to come to the conclusion that purine metabolism facilitates the radiological progression of AS through paracrine signaling. The enhanced level of purine metabolites might be secondary to the pathogenesis of AS, not the cause. The authors should provide more evidence (or at least cite some previous papers) to support their conclusion.

Response: Thank you for your comment. Based on the discovery of mechanism of pathological new bone formation and new treatment for arresting bony fusion in AS, we removed the results of paracrine pathway mediated by purine in CD4-CKO mice in the revised manuscript. We still very appreciate all your comments about purine, which will assist our subsequent study.

5. It has been reported that Shp2 deficient in CD4+ Cells causes cartilage malfunction without affecting T Cell development and functions. ("Ptpn11 Deletion in CD4+ Cells Does Not Affect T Cell Development and Functions but Causes Cartilage Tumors in a T Cell-Independent Manner. Front. Immunol.2017"). The authors should put more discussion on this and describe how this study relates to your present work.

Response: Thank you for your comment. we have cited these prior work and discussed in the revised manuscript (cited in page 9 and page16).

REVIEWERS' COMMENTS

Reviewer #1 (Remarks to the Author):

Most of my concerns have been addressed by authors and the revised manuscript has been notably improved.

Response: Thank you so much for the reviewers' precious comments and suggestions.

Reviewer #2 (Remarks to the Author):

The revisions substantially improve the manuscript and the writing is easier to follow.

Response: Thank you so much for the reviewers' precious comments and suggestions.

Reviewer #3 (Remarks to the Author):

In this revised ms, authors have satisfactorily addressed most of my concerns.

Response: Thank you so much for the reviewers' precious comments and suggestions.